# Association with TFIIIC limits MYCN localisation in hubs of active promoters and chromatin accumulation of non-phosphorylated RNA polymerase II

Raphael Vidal[1,2], Eoin Leen[3], Steffi Herold[1], Mareike Müller[1,4], Daniel Fleischhauer[1], Christina Schülein-Völk[5], Dimitrios Papadopoulos[1,4], Isabelle Röschert[1], Leonie Uhl[1], Carsten P Ade[1], Peter Gallant[1], Richard Bayliss[3], Martin Eilers[1,2]*, Gabriele Büchel[1,2,4]*

[1]Theodor Boveri Institute, Department of Biochemistry and Molecular Biology, Biocenter, University of Würzburg, Würzburg, Germany; [2]Comprehensive Cancer Center Mainfranken, Würzburg, Germany; [3]Astbury Centre for Structural Molecular Biology, Faculty of Biological Sciences, University of Leeds, Leeds, United Kingdom; [4]Mildred Scheel Early Career Center, University Hospital Würzburg, Würzburg, Germany; [5]Theodor Boveri Institute, Core Unit High-Content Microscopy, Biocenter, University of Würzburg, Würzburg, Germany

*For correspondence:
martin.eilers@uni-wuerzburg.
de (ME);
gabriele.buechel@uni-
wuerzburg.de (GB)

**Abstract** MYC family oncoproteins regulate the expression of a large number of genes and broadly stimulate elongation by RNA polymerase II (RNAPII). While the factors that control the chromatin association of MYC proteins are well understood, much less is known about how inter-acting proteins mediate MYC's effects on transcription. Here, we show that TFIIIC, an architectural protein complex that controls the three-dimensional chromatin organisation at its target sites, binds directly to the amino-terminal transcriptional regulatory domain of MYCN. Surprisingly, TFIIIC has no discernible role in MYCN-dependent gene expression and transcription elongation. Instead, MYCN and TFIIIC preferentially bind to promoters with paused RNAPII and globally limit the accumulation of non-phosphorylated RNAPII at promoters. Consistent with its ubiquitous role in transcription, MYCN broadly participates in hubs of active promoters. Depletion of TFIIIC further increases MYCN localisation to these hubs. This increase correlates with a failure of the nuclear exosome and BRCA1, both of which are involved in nascent RNA degradation, to localise to active promoters. Our data suggest that MYCN and TFIIIC exert an censoring function in early transcription that limits promoter accumulation of inactive RNAPII and facilitates promoter-proximal degradation of nascent RNA.

## eLife assessment

This study presents the **valuable** finding that TFIIIC interacts with MYCN to regulate RNA polymerase II dynamics by dissecting its impact on 3D chromatin architecture. Authors provide **convincing** evidence that MYCN and TFIIIC show long-range chromatin contacts, and that the expression of each protein limits the function of the other. The notion emerges that TFIIIC helps MYCN to maintain output at promoters while decreasing less productive associations at larger more extensively connected chromatin hubs. The paper is of interest to molecular biologists working on MYCN-dependent regulation of gene expression.

## Introduction

The MYC family of proto-oncogenes is at the epicentre of cellular regulatory networks that govern cell growth, proliferation, and differentiation (*Dang, 2012*; *Kress et al., 2015*). The three MYC paralogs (MYC, MYCN, and MYCL) are central players in normal development and tissue homeostasis, and when dysregulated, fuel many of the processes that are hallmarks of cancer (*Dhanasekaran et al., 2022*; *Hanahan, 2022*). Among them, MYCN has attracted attention for its causal role in the development of neuroblastoma and other childhood tumours (*Rickman et al., 2018*).

Both MYC and MYCN bind to virtually all active promotors and profoundly alter the dynamics of RNA polymerase II (RNAPII) transcription, with an increase in pause release and elongation being most apparent (*Herold et al., 2019*; *Walz et al., 2014*). One consequence of these changes are alterations in expression of a broad range of target genes (*Dhanasekaran et al., 2023*). Unrelated to those changes in gene expression, MYC and MYCN also control RNAPII function to limit the accumulation of R-loops, to facilitate promoter-proximal double-strand break repair and to coordinate transcription elongation with DNA replication (*Papadopoulos et al., 2023*). Which of these effects are critical for the oncogenic functions of MYC is an open question and consequently the partner proteins via which MYC proteins alter RNAPII dynamics are under intense investigation (*Baluapuri et al., 2019*; *Baluapuri et al., 2020*; *Büchel et al., 2017*; *Das et al., 2023*; *Heidelberger et al., 2018*; *Kalkat et al., 2018*; *Lourenco et al., 2021*; *Oksuz et al., 2023*). Direct interactions of MYC and MYCN with MAX, WDR5, and MIZ1 control the localisation of MYC on chromatin (*Blackwood and Eisenman, 1991*; *Blackwood et al., 1992*; *Thomas et al., 2015*; *Vo et al., 2016*; *Walz et al., 2014*). MYC-dependent effects on RNAPII elongation involves the transfer of elongation factors SPT5 and PAF1c from MYC onto RNAPII (*Baluapuri et al., 2019*; *Endres et al., 2021*; *Jaenicke et al., 2016*). Both MYC and MYCN also interact with and activate topoisomerases I and II, suggesting that MYC/N-dependent pause release also involves the relieve of torsional stress that builds up during early transcription (*Das et al., 2022*).

Intriguingly, both MYC and MYCN form prominent complexes with TFIIIC and mapping experiments for MYCN show that the amino-terminal transcriptional regulatory domain is required for the interaction (*Büchel et al., 2017*; *Heidelberger et al., 2018*). TFIIIC was first identified as a general transcription factor for RNAPIII (*Orioli et al., 2012*). Surprisingly, TFIIIC also binds to thousands of genomic sites that are not shared with RNAPIII and are called extra TFIIIC (ETC) sites (*Noma et al., 2006*). Many ETC sites localise to RNAPII-transcribed promoters and such sites are often juxtaposed to MYCN binding sites (*Büchel et al., 2017*; *Moqtaderi et al., 2010*; *Noma et al., 2006*; *Oler et al., 2010*). TFIIIC is an architectural protein that can affect the three-dimensional chromatin organisation at its binding sites (*Noma et al., 2006*; *Van Bortle and Corces, 2013*). Functionally, TFIIIC can act as insulator that blocks the spread of chromatin states (*Raab et al., 2012*), affect cohesin loading (*Büchel et al., 2017*), and bind at the borders of topologically associating domains (TADs) (*Van Bortle et al., 2014*).

Intriguingly, TFIIIC compartmentalises RNAPII promoters and gene expression. Specifically, the effects of TFIIIC on the expression of E2F-dependent and neuronal genes correlate with its effects on the three-dimensional chromatin architecture. In the case of neuronal genes, TFIIIC prevents the localisation of activity-dependent genes to sites of active transcription before stimulation (*Crepaldi et al., 2013*; *Ferrari et al., 2020*; *Policarpi et al., 2017*). Similarly, TFIIIC together with the activity-dependent neuroprotector homeobox protein controls the three-dimensional architecture of cell cycle genes (*Ferrari et al., 2020*). Our previous experiments had suggested that TFIIIC may have a role in MYCN-dependent control of RNAPII function and we here show that MYCN and TFIIIC together have an unexpected censoring function that excludes non-functional RNAPII from hubs of active promoters.

## Results

### TFIIIC interacts directly with the amino-terminus of MYCN

TFIIIC is present in MYCN immunoprecipitates and the amino-terminal region of recombinant MYCN binds to TFIIIC in pull-down experiments performed with cell lysates (*Büchel et al., 2017*). To test whether MYCN and TFIIIC interact directly, we expressed the six subunits of TFIIIC together with a FLAG-tagged MYCN construct comprising amino acids 2–137 using a recombinant baculovirus

to infect insect cells. Performing pull-down experiments with cell lysates and immobilised FLAG-tagged MYCN, we recovered all six subunits of the TFIIIC complex in the eluate along with the FLAG-tagged MYCN construct, demonstrating that TFIIIC can bind to the amino-terminal region of MYCN (*Figure 1A*). TFIIIC is composed of two subcomplexes, designated TauA ($\tau$A) and TauB ($\tau$B), each comprising three of the six subunits. We first attempted co-expression of FLAG-MYCN with individual subcomplexes, but the $\tau$B complex was too unstable to be isolated, and so we focused on the stable $\tau$A complex. To test whether MYCN binds directly to $\tau$A, we purified MYCN and the $\tau$A complex separately (*Figure 1—figure supplement 1A and B*) and then mixed the purified preparations (*Figure 1B–D*). In gel filtration experiments, the isolated MYCN protein eluted with a molecular weight of around 50 kDa, whereas a fraction of MYCN eluted with a much larger molecular weight together with the $\tau$A complex when both were mixed (*Figure 1B–D*). Native mass spectrometry (*Tamara et al., 2022*) showed the molecular weight of this complex to be 204.7 kDa, very close to the sum of the molecular weights of a 1:1:1:1 complex (*Figure 1—figure supplement 1C and D*). We concluded that the MYCN amino-terminal region and the $\tau$A subcomplex of TFIIIC form a stable complex with each other in solution.

As described in the Introduction, TFIIIC is both a general transcription factor for RNAPIII and has been implicated in regulation of RNAPII-dependent transcription. To test whether TFIIIC is required for the proliferation of neuroblastoma cells, we individually depleted three of its subunits, TFIIIC2, TFIIIC3, and TFIIIC5, by stable expression of doxycycline (Dox)-inducible shRNAs. Controls confirmed that each shRNA efficiently depleted its target protein (*Figure 1—figure supplement 1E*). To test both for a general requirement of each subunit in neuroblastoma cell proliferation and for MYCN-specific effects, we expressed each shRNA in SH-EP-MYCN-ER cell. SH-EP cells express endogenous MYC and are engineered to stably express an MYCN-ER chimera (*Herold et al., 2019*). Activation of the MYCN-ER by addition of 4-hydroxytamoxifen (4-OHT) suppresses the expression of endogenous MYC (*Figure 1—figure supplement 1F*), in effect causing a switch from MYC to MYCN expression. Under control conditions, depletion of TFIIIC2 or TFIIIC3 attenuated proliferation, whereas depletion of the small TFIIIC5 subunit had little effect. Addition of 4-OHT had little effect by itself, but greatly enhanced the inhibitory effect of depletion of each subunit (*Figure 1E*), suggesting that TFIIIC has both general and MYCN-specific roles in neuroblastoma cell proliferation.

## TFIIIC limits promoter binding of non-phosphorylated RNAPII in MYCN-expressing cells

The proximity of TFIIIC binding sites to MYCN binding sites at many promoters transcribed by RNAPII prompted us to explore the role of TFIIIC in MYCN-driven transcription (*Büchel et al., 2017*). Using an antibody that detects total RNAPII, we had previously shown that activation of MYCN in SH-EP-MYCN-ER cells impacts RNAPII in two ways: First, MYCN promotes pause release and MYCN's effect on elongation correlates with its effects on gene expression (*Herold et al., 2019*; *Rahl et al., 2010*; *Walz et al., 2014*). At the same time, activation of MYCN uniformly reduces RNAPII occupancy at all active promoters; this correlates with a reduced accumulation of promoter-proximal R-loops, suggesting that MYCN limits the accumulation of RNAPII at promoters that is impaired in productive elongation and/or splicing (*Herold et al., 2019*).

Previous experiments using antibodies directed against total RNAPII had yielded results that varied among different antibodies (*Büchel et al., 2017* and RV, unpublished). To directly compare the effects of MYCN and TFIIIC on promoter-bound RNAPII with those on elongating RNAPII in a side-by-side manner, we conducted parallel ChIP-sequencing (ChIP-seq) analyses using antibodies that specifically recognise non-phosphorylated and Ser2-phosphorylated (pSer2) RNAPII, respectively. Visual inspection of multiple individual genes (*Figure 2A*) and global analyses (*Figure 2B and C* and *Figure 2—figure supplement 1A and B*) showed that activation of MYCN caused a global decrease in promoter association of non-phosphorylated RNAPII. A control ChIP-seq experiment established that addition of 4-OHT had no such effect in SH-EP cells that do not express an MYCN-ER chimera, demonstrating that the reduction is due to activation of MYCN (*Figure 2—figure supplement 1C*). At the same time, MYCN promoted an increase in transcription elongation as best documented by pSer2-RNAPII occupancy at the transcription end site (TES) (*Figure 2A and C*). This increase was most pronounced on MYCN-activated genes. Comparison with RNA-sequencing (RNA-seq) data showed that it correlated closely with MYCN-dependent changes in gene expression (*Figure 2D*).

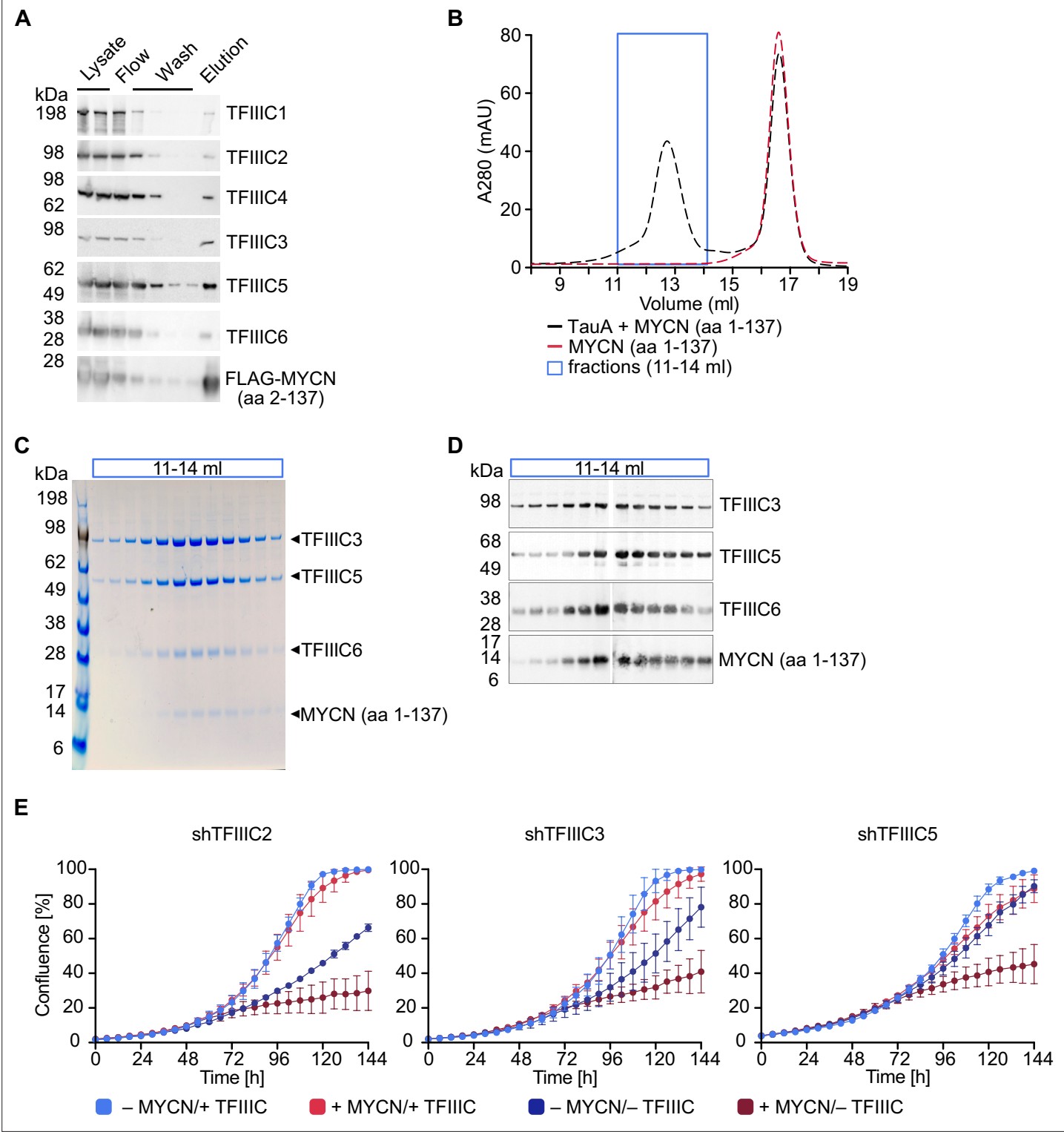

**Figure 1.** TFIIIC directly interacts with MYCN. (**A**) Immunoblots showing levels of FLAG-tagged MYCN (amino acids [aa] 2–137) and the six subunits of the TFIIIC complex after a pull-down assay using anti-FLAG affinity columns. Multiple columns labelled 'Wash' represent the sequential washings (n=2). (**B**) Size exclusion chromatography graph of MYCN (aa 1–137)/TauA (τ A) (black trace) or MYCN alone (red trace). The blue box marks the fractions used for panels C and D (n=2). (**C**) Coomassie staining of fractions of the MYCN (aa 1–137)/τ A complex (fractions marked with blue box in panel B). (**D**) Immunoblot of fractions of the MYCN (aa 1–137)/τ A complex (fractions marked with blue box in panel B). (**E**) Growth curve (measured

*Figure 1 continued on next page*

*Figure 1 continued*

as % confluence) of SH-EP-MYCN-ER cells expressing doxycycline (Dox)-inducible shRNA targeting *TFIIIC2, TFIIIC3,* or *TFIIIC5* under the indicated conditions. Data show mean ± standard deviation (SD) (n=3).

The online version of this article includes the following source data and figure supplement(s) for figure 1:

**Source data 1.** Raw unedited gels for *Figure 1A and D*.

**Source data 2.** Uncropped and labelled gels for *Figure 1A and D*.

**Source data 3.** Raw unedited Coomassie images for *Figure 1C*.

**Source data 4.** Uncropped and labelled Coomassie images for *Figure 1C*.

**Source data 5.** Raw data for graphs shown in *Figure 1B and E*.

**Figure supplement 1.** Characterisation of MYCN/TFIIIC complexes.

**Figure supplement 1—source data 1.** Raw unedited Coomassie images for *Figure 1—figure supplement 1A and B*.

**Figure supplement 1—source data 2.** Uncropped and labelled Coomassie images for *Figure 1—figure supplement 1A and B*.

**Figure supplement 1—source data 3.** Raw data for graphs shown in *Figure 1—figure supplement 1D*.

**Figure supplement 1—source data 4.** Raw unedited gels for *Figure 1—figure supplement 1E and F*.

**Figure supplement 1—source data 5.** Uncropped and labelled gels for *Figure 1—figure supplement 1E, F*.

Depletion of TFIIIC5 abrogated the MYCN-dependent decrease in chromatin association of non-phosphorylated RNAPII but had no effect on MYCN-dependent changes in transcription elongation (*Figure 2A–C*). In many experimental systems, MYC and MYCN effects on elongation parallel closely with the corresponding changes in gene expression. Consistent with these observations, RNA-seq showed that neither depletion of TFIIIC3 nor of TFIIIC5 had significant effects on either basal gene expression or MYCN-dependent changes in steady-state mRNA levels (*Figure 2E* and *Figure 2—figure supplement 1D*). Finally, we aimed to understand whether the localisation of TFIIIC and MYCN at promoters is linked to the dynamics of RNAPII at promoters and performed ChIP-seq of TFIIIC5 in SH-EP-MYCN-ER cells in the presence of 5,6-dichlorobenzimidazole-1-β-D-ribofuranoside (DRB), which stabilises paused RNAPII and prevents pause release of RNAPII (*Mancebo et al., 1997*). Inspection of individual genes (*Figure 2—figure supplement 1E*) and average density plots of all active promoters (*Figure 2F*) showed that activation of MYCN globally enhanced TFIIIC5 association with regions surrounding active transcription start sites, consistent with previous observations (*Büchel et al., 2017*). Addition of DRB increased TFIIIC5 association with active promoters and the combination of MYCN activation and DRB had a strong additive effect, arguing that MYCN preferentially recruits TFIIIC5 to promoters with paused RNAPII.

## MYCN takes part in three-dimensional networks of active promoters

The effects of TFIIIC on the expression of E2F-dependent and neuronal genes correlate with its effects on the three-dimensional chromatin architecture (see Introduction). This raised the possibility that complex formation of TFIIIC similarly affects three-dimensional chromatin interactions of MYCN. To test this hypothesis, we used phosphorylated linker HiChIP (pLHiChIP), a modification of the HiChIP protocol, which identifies pairs of DNA loci that are brought into close spatial proximity to a specific protein (*Figure 3—figure supplement 1A*; *Mumbach et al., 2016*). We initially performed pLHiChIP for MYCN in SH-EP-MYCN-ER neuroblastoma cells (*Figure 3A*). ChIP experiments measuring the occupancy of multiple promoters bound by both MYCN and MYC showed that the chromatin association of MYCN is much greater than that of MYC, allowing analysis of MYCN function on chromatin with little interference from MYC (*Figure 3—figure supplement 1B*). Visual inspection of the MYCN pLHiChIP showed that thousands of pairs of MYCN binding sites are in close spatial proximity to each other (*Figure 3A*). Appropriate quality controls established the validity of these results: For example, the Hi-C module of pLHiChIP protocol yielded a high percentage of valid interaction pairs (*Figure 3—figure supplement 1C and D*). Relative to the Hi-C input, the MYCN HiChIP is strongly enriched for interactions that connected two MYCN binding sites with each other, confirming the specificity of the signal (*Figure 3—figure supplement 1E*).

The pLHiChIP data analysis revealed a total of 4591 distinct binary interactions involving MYCN (*Figure 3B*). To facilitate a comprehensive functional analysis of MYCN anchor sites, we performed ChIP-seq for RNAPII from SH-EP-MYCN-ER cells and integrated the HiChIP data with these data

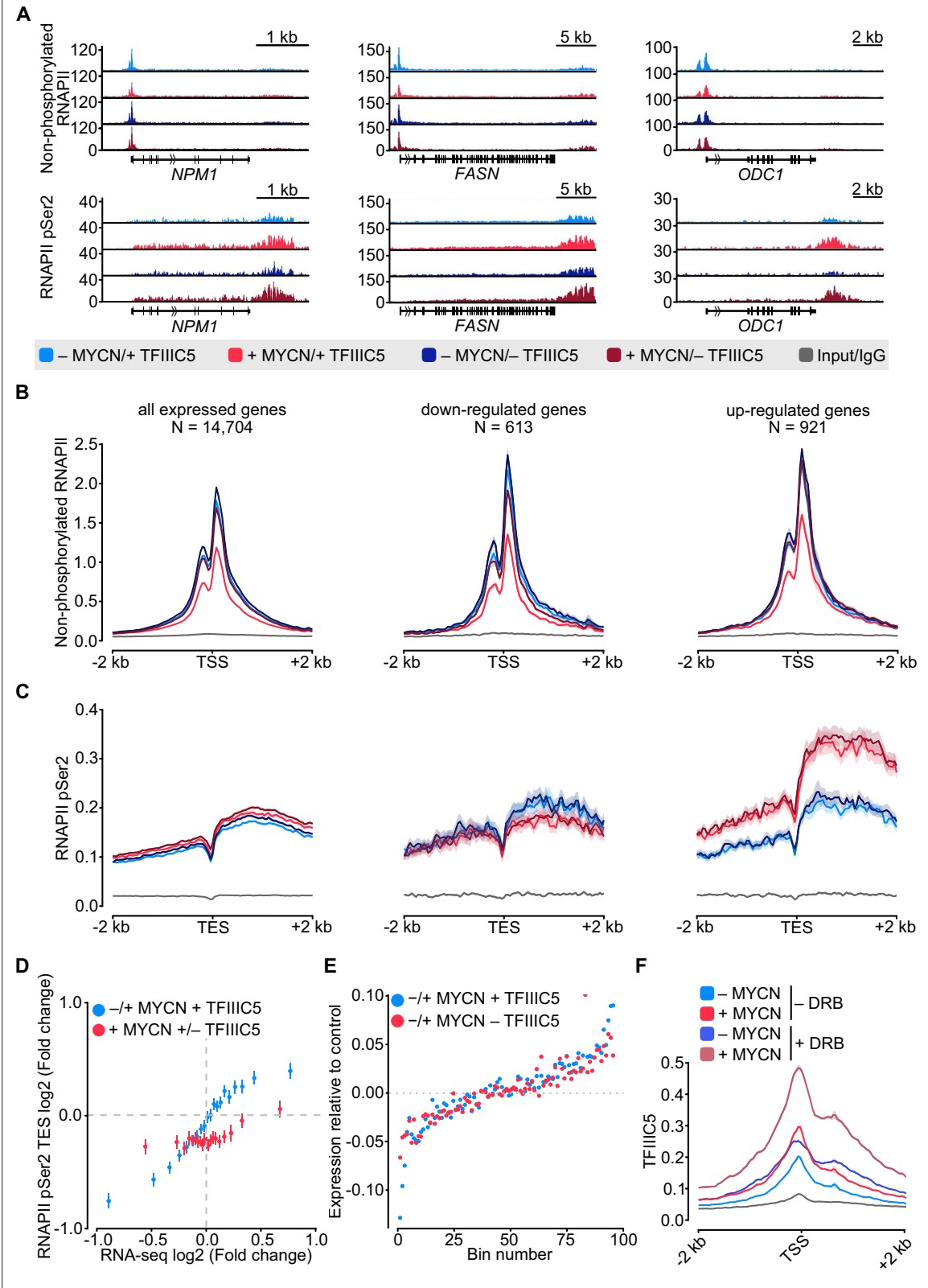

**Figure 2.** MYCN and TFIIIC antagonise accumulation of non-phosphorylated RNA polymerase II (RNAPII). (**A**) Browser tracks for non-phosphorylated RNAPII (top) and RNAPII pSer2 (bottom) ChIP-Rx at the indicated gene loci. SH-EP-MYCN-ER cells were treated with doxycycline (Dox) (1 µg/ml, 48 hr) and/or 4-hydroxytamoxifen (4-OHT), respectively. EtOH was used as control. (**B**) Average density plot of ChIP-Rx signal for non-phosphorylated RNAPII. Data show mean (line) ± standard error of the mean (SEM indicated by the shade) of different gene sets based on an RNA-sequencing (RNA-seq) of

*Figure 2 continued on next page*

*Figure 2 continued*

SH-EP-MYCN-ER cells ± 4-OHT. The y-axis shows the number of spike-in normalised reads and it is centred to the TSS ± 2 kb. N=number of genes in the gene set defined in the Methods (n=2). (**C**) Density plot of ChIP-Rx signal for RNAPII pSer2 as described for panel B. The signal is centred to the transcription end site (TES) ± 2 kb (n=2). (**D**) Average bin dot plot showing fold change for RNAPII pSer2 ChIP-Rx reads over TES ± 2 kb and RNA-seq of SH-EP-MYCN-ER for the same genes ± MYCN + TFIIIC5 (blue) or + MYCN ± TFIIIC5 (red). The plot shows 20 bins representing a total of 13,239 and 12,330 genes for ± MYCN + TFIIIC5 and + MYCN ± TFIIIC5 datasets, respectively (n=3 for RNA-seq, n=2 pSer2 RNAPII ChIP-Rx). (**E**) Average bin dot plot for RNA-seq of SH-EP-MYCN-ER showing log2 mRNA expression normalised by control per bin. Cells were treated with 1 µg/ml Dox ('– TFIIIC5', 48 hr) and/or 4-OHT ('+MYCN', 4 hr) or EtOH as control. Expression was normalised by its control. Each bin represents 150 genes of a total of 14,085 genes. Dotted line marks the relative expression at 0 (n=3). (**F**) Density plot of ChIP-Rx signal for TFIIIC5. Data show mean (line) ± SEM (shade) for 14,722 genes. The signal is centred to the TSS ± 2 kb (n=2).

The online version of this article includes the following figure supplement(s) for figure 2:

**Figure supplement 1.** Effects of MYCN and TFIIIC on RNA polymerase II (RNAPII).

---

(*Figure 3A*), along with other relevant annotations, such as SINE elements. Remarkably, our analysis revealed that 1277 out of the 4591 MYCN anchors (28%) were positioned on RNAPII promoters (*Figure 3B*). Furthermore, 864 out of the 4591 MYCN interactions (19%) exhibited connections between promoters and either exonic or intronic sequences, consistent with previous observations showing that MYCN binding sites are often located within transcribed regions (*Büchel et al., 2017*). Additionally, MYCN interactions were observed between promoters and SINE repetitive elements (647/4591; 14%) including interactions with tRNAs (*Figure 3—figure supplement 1F*). These connections are likely shared with TFIIIC (see below). We also observed interactions between promoters or gene bodies and enhancers, accounting for 453 out of the 4591 interactions (10%).

We used the RNAPII and published MYCN (*Herold et al., 2019*) ChIP-seq data as well as sequencing datasets of mRNA (*Büchel et al., 2017*) and nascent (4sU-labelled) (*Papadopoulos et al., 2022*) RNA to identify the specific properties of promoters within three-dimensional MYCN interactions. This showed that significantly more MYCN and RNAPII bound to promoters that participated in such interactions and that these promoters were more active than MYCN-bound promoters without such interactions, although the latter difference was small *Figure 3C*). We next used the binary interactions as a starting point to reconstruct interaction networks. This showed that MYCN participated in three-dimensional promoter hubs of different sizes (*Figure 3D*), with the largest connecting up to 34 promoters and 10 enhancers (*Figure 3E*). This number of promoters is consistent with published estimates (*Palacio and Taatjes, 2022*). Functional annotation of the promoters that are contained in these large hubs showed that they are highly enriched in genes encoding ribosomal proteins and ribosome biogenesis genes transcribed by RNAPII (*Figure 3—figure supplement 1G*). We concluded from the data that MYCN participates in three-dimensional hubs of highly active promoters; we will use the term 'promoter hubs' (*Lim and Levine, 2021*) for these structures.

## Binding to TFIIIC antagonises MYCN localisation in promoter hubs

To determine whether TFIIIC participates in three-dimensional chromatin structures, we performed pLHiChIP using a previously validated antibody against TFIIIC5 (*Büchel et al., 2017*). In these experiments, thousands of pairs of TFIIIC binding sites yielded a robust signal, demonstrating that they are in close spatial proximity with each other (*Figure 4A*). Specifically, these analyses identified a total of 3499 binary non-contiguous interactions for TFIIIC. Comparison to RNAPII ChIP-seq data showed that 31% of these interactions connected a promoter or the gene body of an RNAPII-transcribed gene to a SINE repetitive element (*Figure 4B*). Importantly, 1107 out of 4591 MYCN interactions shared one (1075) or both (32) anchors with TFIIIC-containing interactions arguing that both proteins participate in common hubs (*Figure 4C*). This is consistent with our previous data, which showed that many promoters contain a TFIIIC-bound site close to an MYCN-bound E-box element (*Büchel et al., 2017*). A significant percentage of the interactions that are shared between MYCN and TFIIIC contained an E-box in one anchor and A- or B-boxes, which are bound by TFIIIC, in the other anchor sequence, arguing that MYCN and TFIIIC can participate in the same interactions (*Figure 4—figure supplement 1A*). Most interactions that are shared by MYCN and TFIIIC connected promoters to other promoters, to gene bodies or to SINE elements. Conversely, TFIIIC5 anchors without overlapping MYCN loops ('TFIIIC5 only') did not show an enrichment for promoter interactions (*Figure 4D*). We concluded that both MYCN and TFIIIC5 are present in promoter hubs. To determine how TFIIIC

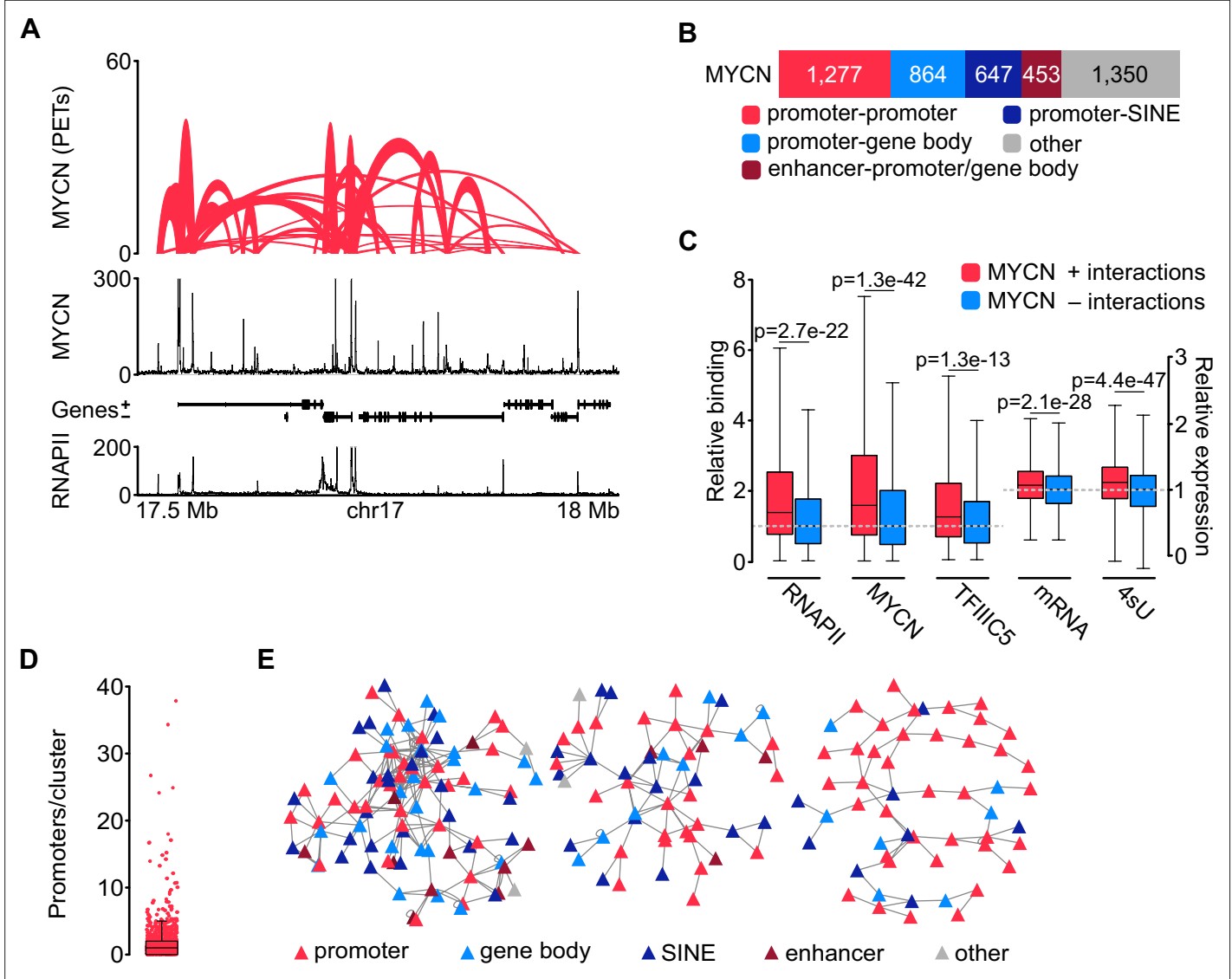

**Figure 3.** MYCN is part of three-dimensional promoter hubs. (**A**) Top: Representative browser track of MYCN three-dimensional chromatin interactions. Height shows the number of paired end tags (PETs) indicating the interaction intensity and the width of the line shows the start and end positions of each anchor. Middle and bottom: Browser tracks showing the number of reads of MYCN and total RNAPII ChIP-Rx, respectively. Unless stated, all experiments were performed in SH-EP-MYCN-ER cells treated with 4-hydroxytamoxifen (4-OHT) (200 nM, 4 hr). The ruler at the bottom shows the genomic coordinates (n=3 independent biological replicates for MYCN phosphorylated linker HiChIP [pLHiChIP]; n=2 for RNAPII ChIP-Rx). (**B**) Bar chart listing functional annotations of all binary MYCN interactions (N=4591; N indicates total number). (**C**) Boxplots showing relative binding of the indicated proteins (RNAPII, MYCN, TFIIIC5) to promoter regions or expression levels of the corresponding genes (mRNA by RNA-sequencing [RNA-seq]; 4sU by 4sU-seq). Red boxes: Genes bound by MYCN and part of MYCN-hubs. Blue boxes: Genes bound by MYCN that are not part of MYCN-hubs. Each pair was normalised to the median of the corresponding 'blue' gene set. p-Values were obtained by pairwise comparisons using Student's t-test (n=2 for TFIIIC5 and RNAPII ChIP-Rx). (**D**) Boxplot showing the number of promoters in each cluster, with each red dot representing one cluster. (**E**) Network reconstruction of the three biggest clusters based on MYCN pLHiChIP interactions. Each anchor is represented by a node ('triangle') and the lines show interactions between the anchors. The colours are indicating the different functional annotation.

The online version of this article includes the following source data and figure supplement(s) for figure 3:

**Figure supplement 1.** Characterisation of HiChip methods.

**Figure supplement 1—source data 1.** Raw data for data shown in *Figure 3—figure supplement 1B*.

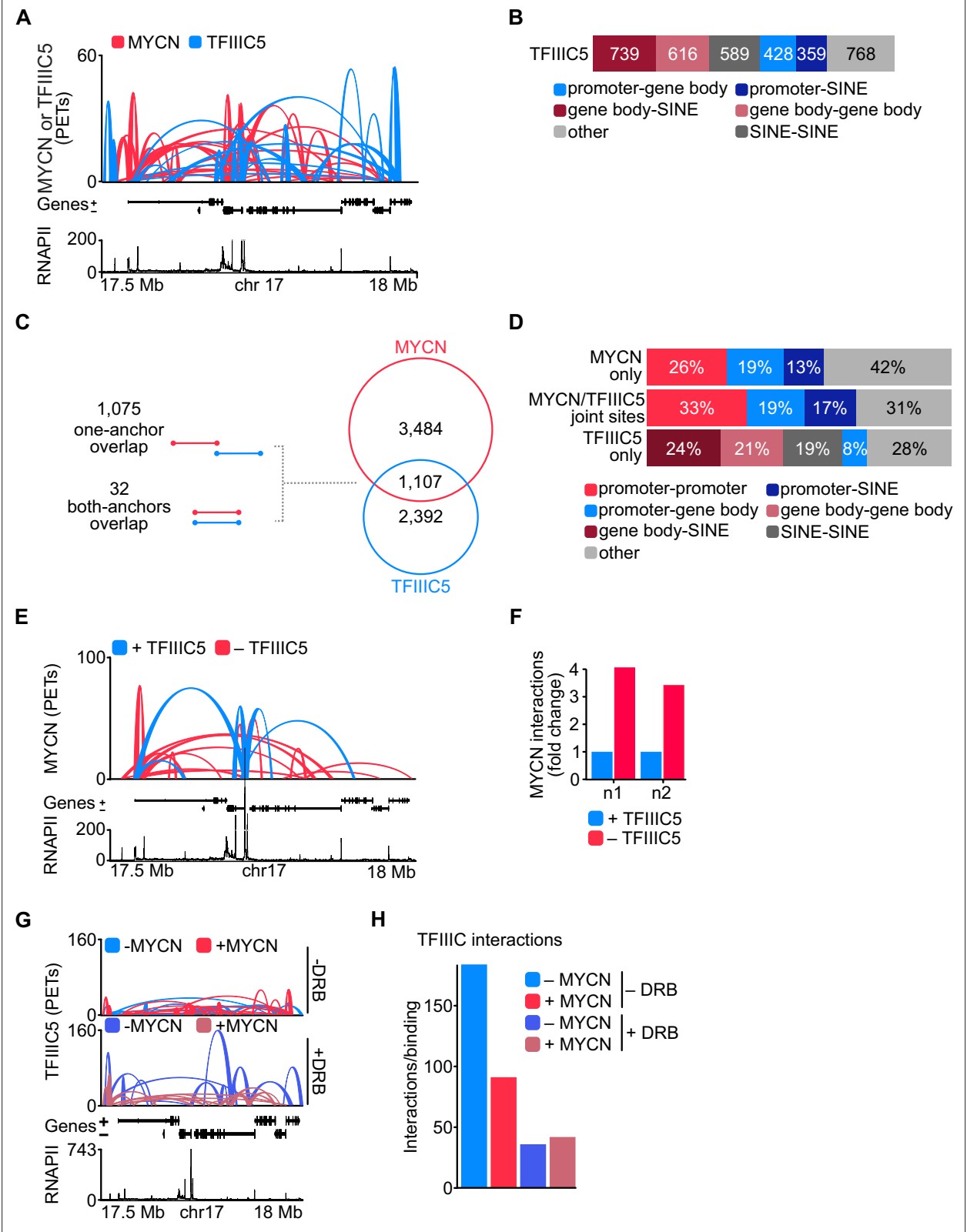

**Figure 4.** TFIIIC antagonises MYCN participation in promoter hubs. (**A**) Representative example of phosphorylated linker HiChIP (pLHiChIP) track for MYCN (red) and TFIIIC5 (blue) interactions (conventions as in *Figure 3A*) (n=2). (**B**) Bar chart listing the total number of functional annotations for all TFIIIC5 binary interactions (N=3499). (**C**) Venn diagram showing the number of interactions shared between MYCN and TFIIIC5. The diagram at the left shows the types of overlaps between connections. (**D**) Bar chart listing the interaction functional annotations for MYCN anchors not overlapping with

*Figure 4 continued*

TFIIIC5 anchors ('MYCN only') as well as TFIIIC5 anchors without overlapping MYCN anchors ('TFIIIC5 only') and their joint anchors. (**E**) Representative example of MYCN spike-in phosphorylated linker HiChIP (spLHiChIP) track for MYCN interactions in the presence (blue) or absence (red) of TFIIIC5 (n=2). (**F**) Bar graph showing the fold change of all MYCN spLHiChIP interactions comparing '+TFIIIC5' and '– TFIIIC5' in SH-EP-MYCN-ER cells expressing a doxycycline (Dox)-inducible shRNA targeting *TFIIIC5*. n1,2 indicates two independent biological replicates. (**G**) Representative example of TFIIIC5 spLHiChIP track without (blue) or with (red) induction of MYCN for SH-EP-MYCN-ER cells (conventions as in *Figure 3A*). (**H**) Bar graph showing the number of TFIIIC5 interactions normalised by the relative binding of TFIIIC5 ChIP-Rx signals for the same coordinates. Coordinates defined as TSS ± 2 kb of 14,722 genes.

The online version of this article includes the following figure supplement(s) for figure 4:

**Figure supplement 1.** Three-dimensional interactions of MYCN and TFIIIC.

affects MYCN involvement in promoter hubs, we used SH-EP-MYCN-ER neuroblastoma cells that express a Dox-inducible shRNA targeting *TFIIIC5* (*Figure 1—figure supplement 1E*) and performed spike-in phosphorylated linker HiChIP (spLHiChIP), a spike-in variation of pLHiChIP that allows quantitative comparisons between different samples. These experiments showed that depletion of TFIIIC5 strongly increased the number of chromatin interactions of MYCN, arguing that association with TFIIIC antagonises MYCN participation in promoter hubs (*Figure 4E and F* and *Figure 4—figure supplement 1B*). HiChIP experiments performed for TFIIIC5 showed that expression of MYCN moderately and blockade of pause release by DRB strongly decreased the frequency of three-dimensional interactions of TFIIIC5 (*Figure 4G and H*). Stratification showed that interactions of TFIIIC5 with promoters, gene bodies, and SINE elements were all decreased upon incubation of cells with DRB (*Figure 4—figure supplement 1C*), arguing that MYCN/TFIIIC5-bound genes are not part of three-dimensional promoter hubs.

## TFIIIC is required for promoter association of factors involved in nascent RNA degradation

Depletion of TFIIIC led to the accumulation of non-phosphorylated RNAPII and an enhanced presence of MYCN in promoter hubs, suggesting that lack of TFIIIC might lead to enhanced or uncontrolled functionality of RNAPII. To ascertain whether TFIIIC is required for functionality of elongating RNAPII, we first performed an rMATS analysis, which analyses changes in splicing (*Shen et al., 2014a*; *Wang et al., 2017*). This showed that depletion of TFIIIC3 caused significant increases in intron retention and exon skipping, evidence of aberrant splicing (*Figure 5—figure supplement 1A and B*). However, most of these alterations occurred in downstream exons, and were not observed for TFIIIC5. Furthermore, expression of MYCN weakened this effect for TFIIIC3 (*Figure 5—figure supplement 1A and B*), arguing that it does account for the requirement for TFIIIC function in MYCN-expressing cells.

To explore promoter-proximal events, we made use of the fact that many of the factors that determine premature termination and degradation of nascent RNA have recently been elucidated and initially used proximity ligation assays (PLAs) to survey a series of factors involved in premature termination (*Rodríguez-Molina et al., 2023*). Appropriate controls using immunofluorescence and PLAs with single antibodies established the specificity of each assay (*Figure 5—figure supplement 1C*). Consistent with the ChIP-seq data, activation of MYCN enhanced the proximity between TFIIIC5 and total RNAPII and depletion of TFIIIC5 reduced the signal, confirming the specificity of the assay (*Figure 5—figure supplement 1D*). Activation of MYCN and depletion of TFIIIC5 had only weak effects on the proximity of RNAPII with NELFE, which associates with pausing RNAPII. Similarly, depletion of TFIIIC5 either alone or in combination with MYCN activation enhanced the proximity of RNAPII with PP2A, which is recruited to promoters via its interaction with the integrator termination complex (*Cossa et al., 2021*; *Vervoort et al., 2021*), and with PNUTS, a targeting subunit of PP1 that is globally involved in termination (*Figure 5A*; *Cortazar et al., 2019*; *Estell et al., 2023*; *Landsverk et al., 2020*).

Key factors in degradation of aberrant nascent RNAs include XRN2, a 5'–3' RNA exonuclease, and the exosome, a 3'–5' exonuclease RNA complex (*Cortazar et al., 2022*; *Gerlach et al., 2022*; *Noe Gonzalez et al., 2021*). We have previously shown that MYCN can recruit the nuclear exosome to its target promoters (*Papadopoulos et al., 2022*). MYCN can also recruit BRCA1, which in turn stimulates binding of the mRNA decapping enzyme DCP1 that initiates RNA degradation (*Herold et al., 2019*). No significant effects were observed in the proximity between RNAPII with the XRN2 exonuclease

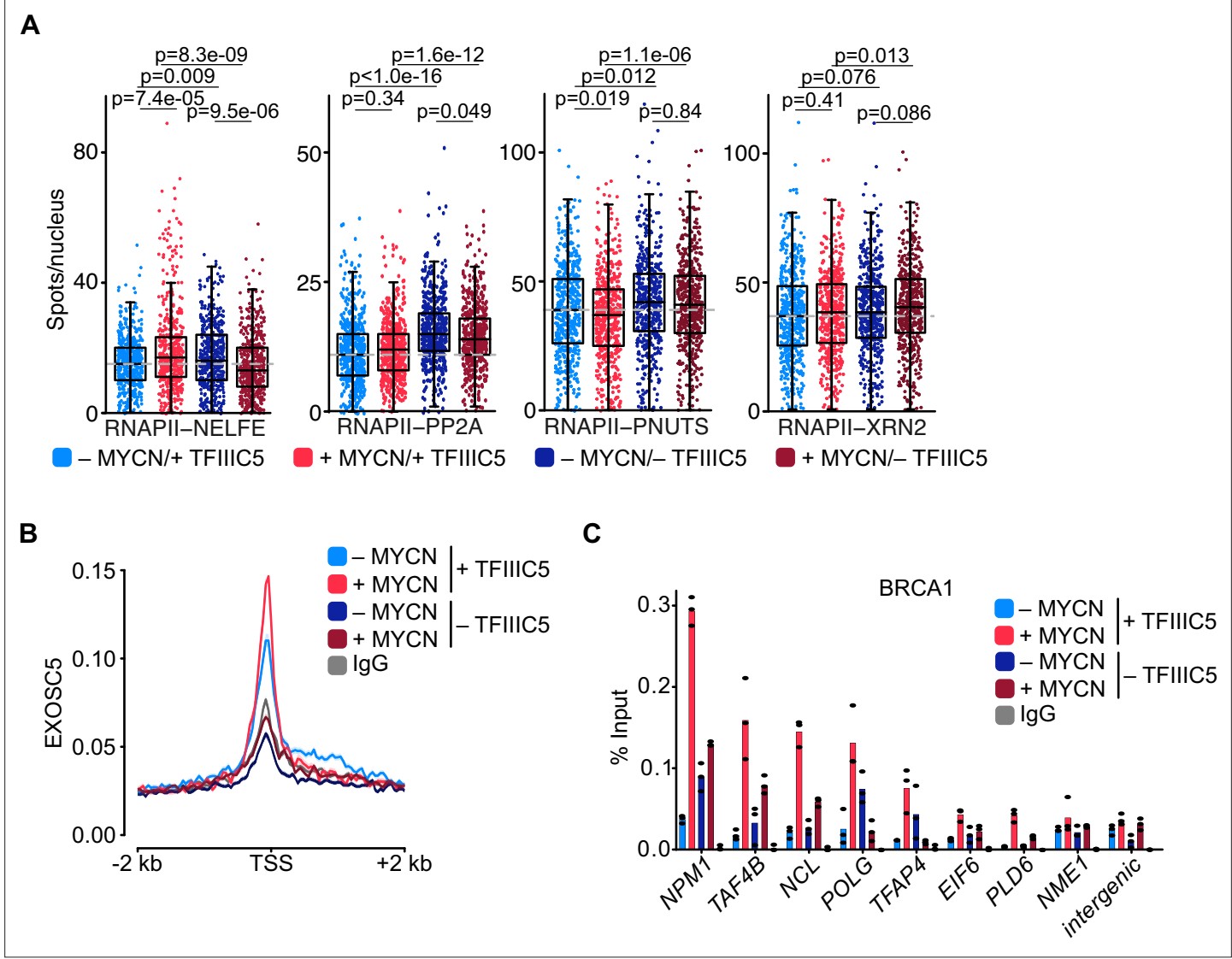

**Figure 5.** TFIIIC is required for promoter association of the exosome and of BRCA1. (**A**) Boxplots showing the number of proximity ligation assay (PLA) signals between RNA polymerase II (RNAPII) and NELFE, PP2A, PNUTS, or XRN2. SH-EP-MYCN-ER cells were treated with 1 µg/ml doxycycline (Dox) ('– TFIIIC5', 48 hr) and/or 4-hydroxytamoxifen (4-OHT) ('+MYCN'). EtOH was used as control. For clarity purposes, 500 cells pooled from different replicates were plotted. p-Values were calculated comparing the PLA signal of all cells using unpaired Wilcoxon rank sum test. The grey dotted line indicates the median in the control condition (n=3). (**B**) Density plot of CUT&RUN for EXOSC5 binding (N=14,704 genes) in SH-EP-MYCN-ER cells expressing a Dox-inducible shRNA targeting *TFIIIC5* treated with 4-OHT. Data show mean ± SEM (shade). (**C**) BRCA1 ChIP in SH-EP-MYCN-ER cells expressing a Dox-inducible shRNA targeting *TFIIIC5* treated with 4-OHT (4 hr). Shown is the mean of technical triplicates of one representative experiment with identical results (n=2).

The online version of this article includes the following source data and figure supplement(s) for figure 5:

**Source data 1.** Raw data for plots shown in *Figure 5A and C*.

**Figure supplement 1.** Effects of TFIIIC on splicing and termination factors.

**Figure supplement 1—source data 1.** Raw data for plots shown in *Figure 5—figure supplement 1D and E*.

(*Figure 5A*). In contrast, CUT&RUN experiments using antibodies-directed EXOSC5, a core structural subunit of the exosome (*Kilchert et al., 2016*), showed that depletion of TFIIIC5 strongly decreased the association of EXOSC5 with regions around the TSS, both in control and in MYCN-expressing cells (*Figure 5B*). To test whether MYCN-dependent recruitment of BRCA1 depends on TFIIIC, we performed ChIP experiments at multiple MYCN-bound promoters (*Figure 5C* and *Figure 5—figure supplement 1E*). Consistent with our previous observations, only low amount of BRCA1 were found

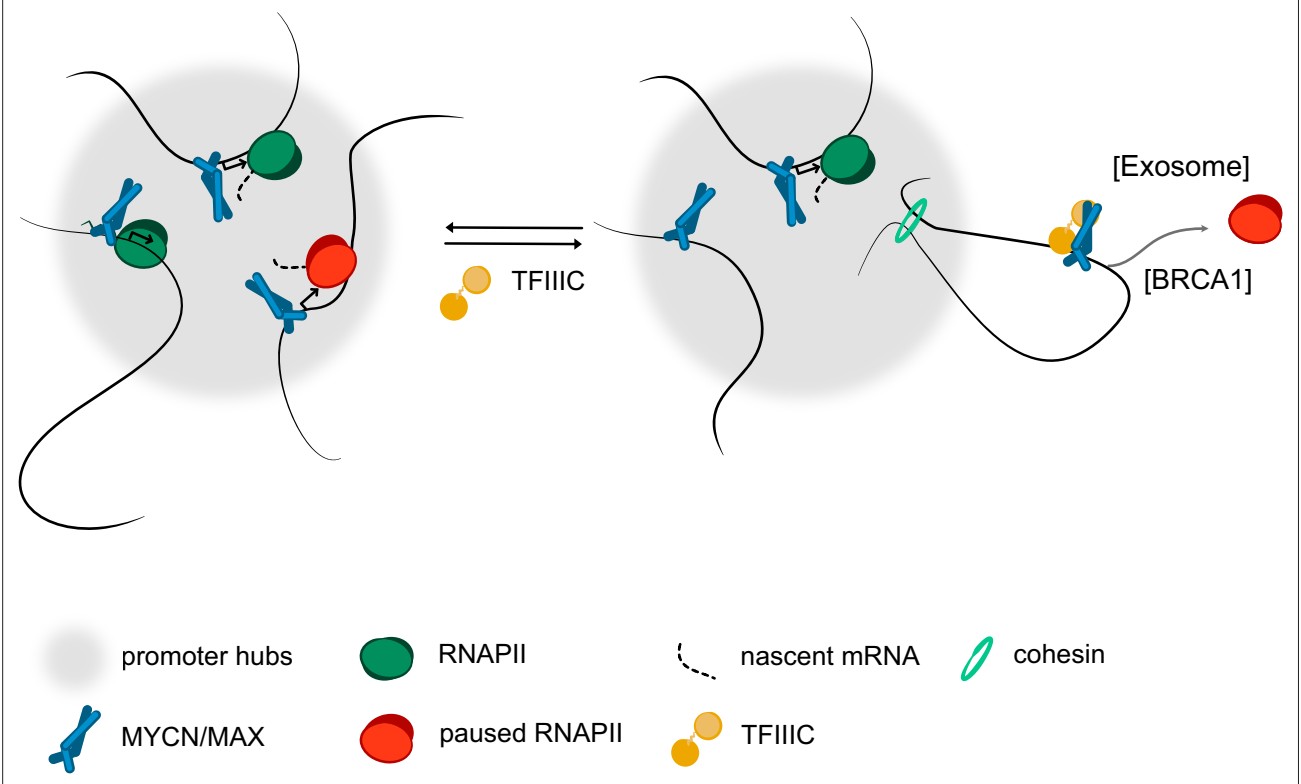

**Figure 6.** Model. Model summarising our findings. We propose that complex formation with the TFIIIC complex antagonise the localisation of MYCN in promoter hubs and that this enables access of the nuclear exosome and BRCA1 to promoters with paused or stalled RNA polymerase II (RNAPII). Both the exosome and BRCA1 have been implicated in fostering the degradation of nascent RNA at promoters. The precise mechanisms by which MYCN and TFIIIC limit accumulation of non-phosphorylated RNAPII at promoters remain to be determined.

at core promoter in SH-EP cells and that induction of MYCN recruited BRCA1 to all tested promoters (*Herold et al., 2019*). Depletion of TFIIIC5 had variable effects on the presence of BRCA1 by itself, but abrogated BRCA1 recruitment by MYCN at all promoters tested (*Figure 5C* and *Figure 5—figure supplement 1E*), demonstrating that TFIIIC5 is required for MYCN-dependent recruitment of BRCA1 to multiple target sites. Collectively, the data show that TFIIIC is required for the association of the nuclear exosome and of BRCA1 with active promoters and that these effects are enhanced in cells expressing MYCN, arguing that they can account for the enhanced dependence of MYCN-expressing cells on TFIIIC. A model summarising our data is shown in *Figure 6*.

## Discussion

One of the most puzzling aspects of MYC biology is the apparent discrepancy between the global association of MYC and MYCN oncoproteins with open chromatin, which includes virtually all active promoters, particularly in tumour cells, and their much more restricted and often weak effects on the expression of downstream target genes. Most MYC/N binding sites on chromatin do not appear to be functional with respect to gene expression and transcriptional elongation (*Kress et al., 2015*; *Sabò et al., 2014*; *Walz et al., 2014*). This raises the possibility that such binding sites are indeed non-functional (*Pellanda et al., 2021*). Alternatively, MYCN has functions in transcription that go beyond gene regulation. In support of the later model, we and others have uncovered critical roles of MYC and MYCN in maintaining the genomic stability of tumour cells, e.g., by coordinating transcription with DNA replication and by enabling promoter-proximal double-strand break repair (*Baluapuri et al., 2020*; *Papadopoulos et al., 2023*). This raises the question of which direct interaction partners of MYC and MYCN mediate these effects and what the underlying mechanisms are. Here, we show that TFIIIC binds directly to the amino-terminus of MYCN, and that a MYCN/TFIIIC complex globally limits the accumulation of non-phosphorylated RNAPII at promoters.

Several mechanisms can account for these observations: First, the amino-terminus of MYC proteins is a transcription-activating domain that can interact with cyclin T1 and CDK8 (*Eberhardy and Farnham, 2002*) and potentially other co-activators, hence TFIIIC may block MYCN interaction with co-activators that recruit RNAPII to promoters. Since MYCN/TFIIIC also restricts RNAPII accumulation at promoters that are not activated by MYCN, this is unlikely to be the sole mode of action. Second, MYCN and TFIIIC binding sites are often downstream of the transcription start sites and the three-dimensional structures formed by MYCN and TFIIIC may sterically restrict elongation and favour termination of non-phosphorylated RNAPII. Third, TFIIIC is an architectural protein complex and re-localisation of promoters may be central to the mechanism of MYCN/TFIIIC action. Active promoters coalesce at a discrete number of sites in the eucaryotic nucleus (*Lim and Levine, 2021*; *Palacio and Taatjes, 2022*). Many of the factors involved in basal transcription undergo liquid-liquid phase separation, arguing that transcription initiation takes place in condensate-like structures established by transcriptional regulatory domains and by RNAPII itself (*Hnisz et al., 2017*; *Lewis et al., 2023*; *Lu et al., 2018*). Consistent with their ubiquitous role in transcription, MYC proteins form condensates by themselves and enter condensates formed by the RNAPII-associated mediator complex (*Boija et al., 2018*; *Solvie et al., 2022*; *Yang et al., 2022*). As described in the Introduction, TFIIIC can restrict the localisation of genes to sites of active transcription. We show here that MYCN/TFIIIC complexes, in contrast to MYCN, are not part of hubs of active promoters, leading us to propose that TFIIIC antagonises MYCN participation in promoter hubs and limits the accumulation of paused or stalled RNAPII at promoters (*Figure 6*). Since TFIIIC competes with the Aurora-A kinase for binding of MYCN and Aurora-A in turn promotes elongation by RNAPII, we propose that the competition between both complexes enables transcriptional elongation on promoters that are activated by MYCN (*Büchel et al., 2017*; *Roeschert et al., 2021*).

As described above, depletion of TFIIIC has no discernible role in elongation and changes in polyA+ mRNA levels, raising the question of what the function of the association might be. The fate of nascent RNA is controlled by several interrelated processes leading to splicing and polyadenylation of the full-length transcript on the one hand, and mis-splicing, premature termination and RNA degradation on the other and MYCN has been implicated in both processes (*Noe Gonzalez et al., 2021*; *Papadopoulos et al., 2022*; *Schmid and Jensen, 2019*). Depletion of TFIIIC strongly decreased the association of the nuclear exosome, an RNA exonuclease complex that degrades multiple forms of aberrant nascent RNA (*Schmid and Jensen, 2019*), and of BRCA1, which in turn recruits the mRNA decapping enzyme DCP1 at promoters (*Herold et al., 2019*). The association of both the exosome and BRCA1 with promoters were enhanced in MYCN-expressing cells, arguing that they are likely to reflect — at least in part – the activity of the TFIIIC/MYCN complex. At the same time, TFIIIC3 depletion caused splicing errors, which occurred predominantly in downstream exons and were independent of MYCN, arguing that the effect on splicing may reflect an activity of TFIIIC that is independent of its function at promoters. We propose that the TFIIIC/MYCN complex formation exerts a censoring and quality control function for RNAPII at promoters that have not received a full complement of activating signals and hypothesise that this contributes to focus the transcription machinery and metabolic resources on the genes that drive the growth of MYCN-driven tumours.

## Methods

Materials used in the study (genetically modified strains, cell lines, reagents, and software) are summarised in the Key resources table in the appendix.

## Cell culture

Cell line derived from human neuroblastoma (SH-EP; CVCL_RR78) was verified by STR profiling and grown in RPMI-1640 (Thermo Fisher Scientific). Murine neuroblastoma cells (NHO2A) were grown in RPMI-1640. Murine NIH-3T3 (CVCL_0594) cells were grown in DMEM (Thermo Fisher Scientific). Media were supplemented with 10% fetal calf serum (Capricorn Scientific GmbH) and penicillin/streptomycin (Sigma-Aldrich). All cells were routinely tested for mycoplasma contamination. Inhibitors were used in the following concentrations: DRB: 100 μM, 2 hr. Growth curves were obtained using the Incucyte Live-Cell Analysis System.

## Transfection and lentiviral infection

For lentivirus production, HEK293TN (CVCL_UL49) cells were transfected using PEI (polyethylenei-mine, Sigma-Aldrich). Lentiviruses expressing an shRNA targeting *GTFIIIC5* (targeting sequence #1: AAGCGCAGCACCTACAACTACA, #2: TTGATAAATCTTGGCATCTGGG) were produced by transfection of pINDUCER11 (Sequence #1) and pLT3GEPIR (Sequence #2) plasmid together with the packaging plasmid psPAX.2 and the envelope plasmid pMD2.G into HEK293TN cells. Virus-containing supernatant was harvested 24 hr and 48 hr after transfection. SH-EP-MYCN-ER cells were infected with lentiviral supernatants in the presence of 4 µg/ml polybrene (Sigma-Aldrich) for 24 hr. Cells were sorted for GFP and RFP expression. shRNA expression was stimulated by addition of Dox (1 µg/ml) for 12 hr and cells were FACS-sorted for RFP-GFP double-positive cells. Lentiviruses expressing an shRNA targeting *GTFIIIC2 or GTFIIIC3* (targeting sequence *GTFIIIC2:* TGAAGCAGAAGAATGGTCTGGA, *GTFIIIC3:* TTCATCATTTTCTTGGTTTCAC) were produced by transfection of and pLT3GEPIR plasmid together with the packaging plasmid psPAX.2 and the envelope plasmid pMD2.G into HEK293TN cells. After harvesting supernatant and infection SH-EP-MYCN-ER cells were selected using puromycin and sorted for GFP expression.

For experiments, cells were harvested 48 hr after induction with Dox (1 µg/ml) or ethanol as control. For induction of the MYCN chimera cells were treated with 4-OHT (200 nM, 4 hr) as indicated.

## Constructs

All constructs were cloned from human cDNA or sub-cloned from plasmids containing specific human cDNA sequences. All TFIIIC subunit containing complexes (TFIIIC/3xFLAG-MYCN; τA/3xFLAG-MYCN; τA) were expressed using the MultiBac system. For τA/3xFLAG-MYCN, TFIIIC3, TFIIIC5, Strep-TFIIIC6, and 3xFLAG-MYCN aa 2–137 were cloned into pACEBAC1, pIDC, pIDK, and pIDS, respectively. These were sequentially assembled into a single vector using Cre-Lox recombination. For the TFIIIC/3xFLAG MYCN complex, 10xHis-TFIIIC1, TFIIIC3, TFIIIC5, TFIIIC6, and 3xFLAG MYCN aa 2–137 were all cloned into pACEBAC1. A single pACEBAC1 vector containing all of these inserts was generated by repetitive sub-cloning of the subunits. The acceptor pACEBAC1 was opened using I-CeuI and SpeI. The region of interest was removed from the donor pACEBAC1 clone using I-CeuI and AvrII. The region of interest was then ligated into the acceptor pACEBAC1. An internal AvrII site had to be silently removed from TFIIIC3 by site-directed mutagenesis prior to sub-cloning. The pACEBAC1 with these five members was then used in successive Cre-Lox recombination reactions with pIDK TFIIIC4 and then pIDC TFIIIC2 in order to generate a construct with all seven members of the complex. For the τA complex 6xhis-TEV-TFIIIC3 was cloned into pACEBAC1. This was Cre-loxed with pIDK Strep-TFIIIC6 and pIDC TFIIIC5. MultiBac constructs were validated by restriction enzyme digest and sequencing. Post assembly, constructs containing multiple subunits were validated by analytical PCR and restriction enzyme digest.

## Protein expression and purification

For *Sf9* expression, bacmid DNA was generated by transformation of expression vectors into *E. coli* DH10MultiBac cells. Purified bacmid DNA (1.6–8.3 µg) was transfected into a monolayer of *Sf9* cells, in a T25 flask, using X-tremeGENE HP DNA Transfection Reagent (Roche). After 7 days at 28°C the P1 virus stock was collected. 1 ml of P1 virus was used to infect 300 ml of *Sf9* cells at a density of $2\times10^6$ cells/ml. The infected cells were grown for 72 hr at 120 rpm and 27°C prior to harvesting by centrifugation at $1000\times g$ for 10 min. The supernatant was used to infect large-scale cultures of *Sf9* cells. 100 ml of virus stock was used per litre of *Sf9* cells at a density of $1.5\times10^6$ cells/ml. These were harvested after 48 hr. *E. coli* expression was performed using BL21(DE3)RIL cells. 10 ml from overnight cultures were used to seed each litre of LB media (supplemented with 50 µg/ml kanamycin and 35 µg/ml chloramphenical). The cells were grown at 37°C and 220 rpm until mid log phase (~0.6 $OD_{600}$) where upon expression was induced using 0.6 mM final isopropyl β-D-1-thiogalactopyranoside. Cells were grown overnight at 20°C, and were harvested by centrifugation at $6000 \times g$ for 15 min. Cell pellets were stored at –80°C prior to purification.

For the purification of TFIIIC/3xFLAG-MYCN and τA/3xFLAG-MYCN complexes *Sf9* pellets were resuspended in a lysis buffer (25 mM Tris pH 8, 200 mM NaCl, EDTA free cOmplete PI [Roche], 10 µg/ml DNAseI) then lysed by sonication. Lysate was clarified at $40,000 \times g$ for 20 min. Clarified lysate was applied to 2 ml bed volume Anti-FLAG M2 Affinity Gel (Sigma-Aldrich). The slurry was rotated in the

cold room for 2 hr prior to application to a gravity flow column. Flow was collected and resin washed as follows. For the TFIIIC/3xFLAG-MYCN complex the resin was washed with 40 ml of wash buffer (25 mM Tris pH 8, 200 mM NaCl) in four 10 ml batches. For the $\tau$ A/3xFLAG-MYCN complex the resin was washed with two 10 ml batches of wash buffer, followed by 10 ml of wash buffer with a final NaCl concentration of 350 mM, followed by 10 ml of wash buffer with a final NaCl concentration of 400 mM. The resin was finally washed with 10 ml of wash buffer. In both cases the protein was eluted by addition of 17.5 ml wash buffer spiked with a final concentration of 0.25 mg/ml 3xFLAG peptide. For the $\tau$ A/3XFLAG-MYCN complex the eluate fractions were spiked with 2 mM final β-mercaptoethanol and then concentrated to less than 1 ml with a Vivaspin 6 10 kDa MWCO centrifugal concentrator. Size exclusion chromatography (SEC) was performed using a Superose 6 10/300 GL column attached to an ÄKTA pure FPLC system (Cytiva). The SEC buffer (25 mM Tris pH 8, 200 mM NaCl, 3 mM DTT) was flowed over the column at 0.3 ml/min and fractions collected every 0.25 ml over the peaks. In order to do an MYCN alone SEC run, the MYCN peak fractions from several SEC runs of the complex were combined and re-concentrated down to 300 µl. This was then run as previous. For all purified proteins, protein concentration was determined by absorbance at 280 nm. Unless otherwise stated proteins were aliquoted, flash-frozen in liquid $N_2$, and stored at –80°C prior to use.

For the purification of $\tau$ A alone, *Sf9* pellets were resuspended in a base buffer (25 mM Tris pH 7.4, 2.7 mM KCl, 287 mM NaCl, 2 mM β-mercaptoethanol) supplemented with 0.1 mg/ml DNase I, and 1 EDTA free cOmplete PI (Roche) tablet per 30 ml of buffer. Cells were lysed by sonication, then the lysate clarified at 40,000 × *g* for 25 min. Clarified lysate was applied to 2.5 ml bed volume His-select Cobalt Affinity Gel (Sigma-Aldrich). The slurry was rotated in the cold room for an hour prior to application to a gravity flow column. The resin was washed sequentially as follows: 50 ml base buffer, 10 ml ATP buffer (25 mM Tris pH 7.4, 137 mM NaCl, 2.7 mM KCl, 5 mM $MgCl_2$, 5 mM ATP), 50 ml base buffer with 5 mM imidazole, 50 ml base buffer with 10 mM imidazole. Protein was eluted sequentially by addition of 25 ml of base buffer with 150 mM imidazole followed by 10 ml of the same buffer with 500 mM imidazole. Eluted fractions were pooled and 0.7 mg of His-tagged TEV NIa was added. The mixture was dialysed overnight, using 10 kDa MWCO Snakeskin dialysis tubing, against 4 l of buffer (25 mM Tris pH 7.4, 137 mM NaCl, 2.7 mM KCl, 2 mM β-mercaptoethanol). Post dialysis the TEV NIa processed complex was rebound to Cobalt resin as per previous. The flow fraction was collected and concentrated. SEC was performed as per the $\tau$ A/3XFLAG-MYCN complex. $\tau$ A was concentrated, as per pre-SEC, down to 16.8 µM and used for $\tau$ A/MYCN complex reconstruction immediately.

Protein purification was assessed by a combination of Coomassie-stained SDS-PAGE gels and immunoblotting (TFIIIC90: A301-239A, RRID:AB_890667; TFIIIC5: A301-242A, RRID:AB_890669; TFIIIC102: A301-238A, RRID:AB_890671, Bethyl Laboratories; TFIIIC110: sc-81406, RRID:AB_2115237; MYCN: sc-53993, RRID:AB_831602, Santa Cruz Biotechnologies; TFIIIC35: NBP2-31851, RRID:AB_2891101; TFIIIC1: NBP2-14077, RRID:AB_2891102, Novus Biologicals). The SEC elution position of molecular weight standards were established by using Bio-Rad gel filtration standards (cat #1511901). 5 mg/ml BSA was additionally used as the 66 kDa marker. Dextran blue was used to establish the void position of the column.

Aurora-A aa 122–403 D274N C290A C393A was purified as previously described (*Bayliss et al., 2003*). The final buffer was 20 mM HEPES pH 7.5, 150 mM NaCl, 5 mM $MgCl_2$, 10% glycerol, 1 mM β-mercaptoethanol. *E. coli*-expressed MYCN aa 1–137 pellets were initially disrupted using base buffer (pH 7.5, 100 mM $KH_2PO_4$, 10 mM Tris, 300 mM NaCl, 2 mM β-mercaptoethanol) which was supplemented with 30 mg of lysozyme, 0.3 µg DNaseI, and 0.9 µM $MnCl_2$ per 30 ml of base buffer. This was sonicated 15 s on 15 s off for 150 s at 40% amplitude. The lysate was initially clarified by centrifugation for 30 min at 30,000 × *g*. As this construct of MYCN expresses into inclusion bodies, the protein had to be recovered from the pellet using urea. All urea solutions were prepared immediately prior to use. The pellet was disrupted by vigorous pipetting after addition of 30 ml of 1 M urea in base buffer followed by sonication as previous. This solution was clarified by centrifugation for 20 min at 30,000 × *g*. This protocol of disruption of the pellet, following addition of urea containing base buffers, followed by clarification was repeated sequentially for solutions containing 2 M and 4 M urea. 6xHis-tagged MYCN aa 1–137 was found in the soluble fraction of the 4 M urea solution. This was diluted 1/4 with 25 mM Tris pH 7.4, 2.7 mM KCl, 137 mM NaCl, 2 mM β-mercaptoethanol prior to application of the protein to His-Select Cobalt affinity gel (Sigma-Aldrich). The slurry was rotated for 60 min prior to application to a gravity flow column. The resin was washed sequentially as follows with

imidazole containing buffer (25 mM Tris pH 7.4, 2.7 mM KCl, 137 mM NaCl, 2 mM β-mercaptoethanol): 50 ml 0 mM imidazole, 50 ml 5 mM imidazole, 25 ml 10 mM imidazole, 25 ml 150 mM imidazole, and 10 ml 500 mM imidazole. Pure fractions were pooled, along with 0.7 mg of His-tagged TEV NIa, and were dialysed overnight using 3.5 kDa MWCO snakeskin dialysis tubing against 4 l of buffer (25 mM Tris pH 7.4, 2.7 mM KCl, 137 mM NaCl, 2 mM β-mercaptoethanol). Post TEV Nia processing, the protein was re-applied to His-Select Cobalt affinity gel (Sigma-Aldrich). The flowthrough fractions were pooled and concentrated (Vivaspin Turbo 3000 MWCO) prior to SEC. SEC was performed using an ÄKTA prime and 16/600 S75 column. Pure fractions were pooled and concentrated as per pre-SEC. Protein concentration was determined using absorbance at 280 nm.

## τA/3xFLAG-MYCN native mass spectrometry

The τA/3xFLAG-MYCN complex was desalted into 200 mM ammonium acetate using 75 µl Zeba spin desalting columns (Thermo Fisher Scientific). Samples were analysed by nanoelectrospray ionisation MS using a quadrupole-orbitrap MS (Q-Exactive UHMR, Thermo Fisher Scientific) using gold/palladium-coated nanospray tips prepared in-house. The MS was operated in positive ion mode using a capillary voltage of 1.3 kV, capillary temperature of 250°C and S-lens RF of 200 V. In-source trapping was used with a desolvation voltage of −100 V for 4 µs. Extended trapping was not used. The quadrupole mass range was 2000–15,000 m/z. Nitrogen gas was used in the HCD cell with a trap gas pressure setting of 5. Orbitrap resolution was 12,500, detector m/z optimisation was low. Five microscans were averaged. Mass calibration was performed by a separate injection of sodium iodide at a concentration of 2 µg/µl. Data processing was performed using QualBrowser 4.2.28.14 and deconvoluted using UniDec (*Marty et al., 2015*).

## τA/3xFLAG-MYCN intact mass spectrometry

Protein desalting and mass analysis was performed by LC-MS using an M-class ACQUITY UPLC (Waters UK, Manchester, UK) interfaced to a Xevo QToF G2-XS mass spectrometer (Waters UK, Manchester, UK). Samples were diluted to 1 µM using 0.1% TFA. 1 µl of the 1 µM sample was run on an Acquity UPLC Protein BEH C4 column (300 Å, 1.7 µm, 2.1 mm × 100 mm, Waters UK) with an Acquity UPLC Protein BEH VanGuard Pre-Column (300 Å, 1.7 µm, 2.1 mm × 5 mm, Waters UK). Buffer A was 0.1% formic acid in water, and buffer B was 0.1% formic acid in AcN (vol/vol basis). System flow rate was kept constant at 50 µl/min. Protein sample was loaded on to the trap column in 20% acetonitrile/0.1% formic acid and washed for 5 min. Following valve switching, the bound protein was eluted by a gradient of 20–95% solvent B in A over 10 min. The column was subsequently washed with 95% solvent B in A for 5 min before re-equilibration at 20% solvent B in A ready for the next injection. The mass spectrometer was calibrated using a separate injection of glu-fibrinopeptide. Data were processed using MassLynx 4.2.

## Protein identification/peptide LC-MS analysis

Gel bands were excised and chopped into small pieces (~1 mm³), covered with 30% ethanol in a 1.5 ml microcentrifuge tube and heated to 56°C for 30 min with shaking. The supernatant was removed, replaced with fresh ethanol solution, and was again heated to 56°C for 30 min with shaking. This was repeated until all Coomassie stain was removed from the gel. The gel slices were then dehydrated by covering with 100% acetonitrile and left for 5 min before the supernatant was discarded and replaced with a fresh aliquot of acetonitrile. Proteins were reduced by adding 100 µl 20 mM DTT solution before incubation at 57°C for 1 hr with shaking. The supernatant was removed and once the gel pieces were at room temperature (RT) proteins were alkylated by adding 100 µl 55 mM iodoacetic acid. The samples were then incubated at RT in the dark for 30 min with shaking. After removing the supernatant, the gel slices were then covered with 100% acetonitrile and left for 5 min. The acetonitrile was removed, and the gel pieces were left to dry in a laminar flow hood for 60 min. Once dry, the gel slices were cooled on ice then they were covered with ice-cold protease solution and left on ice for 20 min to rehydrate. Trypsin solution (20 ng/µl in 25 mM ammonium bicarbonate) was added to the unknown bands, chymotrypsin solution (25 ng/µl in 25 mM ammonium bicarbonate). Excess protease solution was removed, and the gel slices were covered with a minimal amount of 25 mM ammonium bicarbonate. After briefly vortexing and centrifuging, the gel slices were incubated at 37°C while shaking for 18 hr. The resulting digest was vortexed and centrifuged. The supernatant was

recovered and added to a 1.5 ml tube containing 5 μl acetonitrile/water/formic acid (60/35/5; vol/vol). 50 μl acetonitrile/water/formic acid (60/35/5; vol/vol) was added to the gel slices and vortexed for an additional 10 min. The supernatant was pooled with the previous wash and one additional wash of the gel slices was performed. The pool of three washes was dried by vacuum centrifugation. The peptides were reconstituted in 20 μl 0.1% aqueous trifluoroacetic acid prior to analysis. 4 μl sample were injected onto an in-house-packed 20 cm capillary column (inner diameter 75 μm, 3.5 μm Kromasil C18 media). An EasyLC nanoliquid chromatography system was used to apply a gradient of 4–40% ACN in 0.1% formic acid over 45 min at a flow rate of 250 nl/min. Total acquisition time was 60 min including column wash and re-equilibration. Separated peptides were eluted directly from the column and sprayed into an Orbitrap Velos Mass Spectrometer (Thermo Fisher Scientific, Hemel Hempstead, UK) using an electrospray capillary voltage of 2.7 kV. Precursor ion scans were acquired in the Orbitrap with resolution of 60,000. Up to 20 ions per precursor scan were selected for fragmentation in the iontrap. Dynamic exclusion of 30 s was used. Peptide MS/MS data were processed with PEAKS Studio X+ (Bioinformatic Solutions Inc, Waterloo, Ontario, Canada) and searched against the Uniprot databases for *Homo sapiens* and *S. frugiperda* proteins (release 2021_02). Carbamiodomethylation was selected as a fixed modification, variable modifications were set for oxidation of methionine and deamidation of glutamine and asparagine. MS mass tolerance was 15 ppm, and fragment ion mass tolerance was 0.3 Da. The peptide false discovery (FDR) rate was set to 1%.

## Immunoblots

Whole-cell extracts were prepared using RIPA buffer (50 mM HEPES pH 7.9, 140 mM NaCl, 1 mM EDTA; 1% Triton X-100, 0.1% sodium deoxycholate, 0.1% SDS) containing protease and phosphatase inhibitor cocktails (Sigma-Aldrich). Lysates were cleared by centrifugation and protein concentrations were determined by Bradford.

Protein samples were separated on Bis-Tris gels and transferred to a PVDF membrane (Millipore). For immunoblots showing multiple proteins with similar molecular weight, one representative loading control is shown. Vinculin (VCL) or GAPDH were used as loading control. Antibodies used in this study: MYCN (B8.4.B): sc-53993, RRID:AB_831602, Santa Cruz Biotechnology; TFIIIC5: A301-242A, RRID:AB_890669, Bethyl Laboratories; MYC (Y69): ab32072, RRID:AB_731658, Abcam; VCL (h-VIN1): V9131, RRID:AB_477629, Sigma-Aldrich; GAPDH: 2118, RRID:AB_561053, Cell Signaling; TFIIIC2: sc-81406, RRID:AB_2115237, Santa Cruz; TFIIIC3: sc-393235, Santa Cruz.

## Proximity ligation assay

SH-EP-MYCN-ER cells were plated in 384-well plates (PerkinElmer), treated with Dox and/or 4-OHT and fixed with methanol for 20 min. After blocking for 30 min with 5% BSA in PBS, cells were incubated overnight at 4°C with primary antibodies: Total RNAPII (F12): sc-55492; TFIIIC5: A301-242A, Bethyl Laboratories; NELFE: ABE48, Merck; PP2A: 2038, Cell Signaling; PNUTS: A300-439-1, Bethyl Laboratories; XRN2: A301-103A, Bethyl Laboratories. PLA was performed using Duolink In Situ Kit (Sigma-Aldrich) according to the manufacturer's protocol. Nuclei were counterstained using Hoechst 33342 (Sigma-Aldrich). Pictures were taken with Z-stacks of four planes at 0.5 μm distance at 40-fold magnification in Operetta CLS High-Content Imaging System. Analysis was performed in Harmony High Content Imaging and Analysis Software. Thirty image fields per well were acquired with a total of at least 500 cells per sample.

## High-throughput sequencing

ChIP and ChIP-seq was performed as described previously (*Roeschert et al., 2021*). For each ChIP or ChIP-Rx seq experiment, $5 \times 10^7$ cells per immunoprecipitation condition were fixed for 5 min at RT with formaldehyde (final concentration, 1%). Fixation was stopped by adding 125 mM glycine for 5 min. Cells were harvested in ice-cold PBS containing protease and phosphatase inhibitors (Sigma-Aldrich). As exogenous control (spike-in), murine NHO2A, or NIH3T3 cells were added at a 1:10 cell ratio during cell lysis. Cell lysis was carried out for 20 min in lysis buffer I (5 mM PIPES pH 8.0, 85 mM KCl, 0.5% NP-40) and nuclei were collected by centrifugation (1500 rpm, 10 min, 4°C). Crosslinked chromatin was prepared in lysis buffer II (10 mM Tris pH 7.5, 150 mM NaCl, 1 mM EDTA, 1% NP-40, 1% sodium deoxycholate, 0.1% SDS) and fragmented by using the Covaris Focused Ultrasonicator M220 for 50 min/ml lysate. For each IP reaction, 15 μl (for ChIP) or 100 μl (for ChIP-seq) Dynabeads Protein

A and Protein G (Thermo Fisher Scientific) were pre-incubated overnight with rotation in the presence of 5 mg/ml BSA and 3 µg (for ChIP) or 10–15 µg (for ChIP-seq) antibody (MYCN [B8.4.B]: sc-53993, RRID:AB_831602; TFIIIC5: A301-242A, RRID:AB_890669; RNAPII pSer2: ab5095, RRID:AB_304749; RNAPII [8WG16]: sc-56767, RRID:AB_785522; BRCA1: A300-000A, RRID:AB_67367, Bethyl Laboratories). Chromatin was added to the beads, and IP was performed for a minimum of 6 hr at 4°C with rotation. Beads were washed three times each with washing buffer I (20 mM Tris pH 8.1, 150 mM NaCl, 2 mM EDTA, 1% Triton X-100, 0.1% SDS), washing buffer II (20 mM Tris pH 8.1, 500 mM NaCl, 2 mM EDTA, 1% Triton X-100, 0.1% SDS), washing buffer III (10 mM Tris pH 8.1, 250 mM LiCl, 1 mM EDTA, 1% NP-40, 1% sodium deoxycholate, including a 5 min incubation with rotation), and TE buffer. Chromatin was eluted twice by incubating with 200 µl elution buffer (100 mM NaHCO$_3$, 1% SDS in TE) for 15 min with rotation. Input samples and eluted samples were de-crosslinked overnight. Protein and RNA were digested with proteinase K and RNase A, respectively. DNA was isolated by phenol-chloroform extraction followed by an ethanol precipitation and analysed by qPCR using StepOnePlus Real-Time PCR System (Thermo Fisher Scientific) and SYBR Green Master Mix (Thermo Fisher Scientific) or sequencing on the Illumina NextSeq 2000.

After DNA extraction occupancy of different proteins were assessed by RT-PCR. Primers were used for *TFAP4* (forward: CCGGGCGCTGTTTACTA; reverse: CAGGACACGGAGAACTACAG), *POLG* (forward: CTTCTCAAGGAGCAGGTGGA; reverse: TCATAACCTCCCTTCGACCG), *NPM1* (forward: TTCACCGGGAAGCATGG; reverse: CACGCGAGGTAAGTCTACG), Intergenic region (forward: TTTT CTCACATTGCCCCTGT; reverse: TCAATGCTGTACCAGGCAAA), *NCL* (forward: CTACCACCCTCA TCTGAATCC; reverse: TTGTCTCGCTGGGAAAGG), *NME1* (forward: GGGGTGGAGAGAAGAA AGCA; reverse: TGGGAGTAGGCAGTCATTCT), *PLD6* (forward: GCTGTGGGTCCCGGATTA; reverse: CCTCCAGAGTCAGAGCCA), *TAF4B* (forward: AAGGTCGTCGCTCACAC, reverse: GCGTGGCTATAT AAACATGGCT), *RPL22* (forward: CCGTAGCTTCCTCTCTGCTC, reverse: ACCTCTTGGGCTTCCT GTCT), *CCND2* (forward: GCCAGCTGCTGTTCTCCTTA, reverse: CCCCTCCTCCTTTCAATCTC). Shown analysis of RT-PCR show mean of technical triplicates as well as an overlay of each data point to indicate the distribution of the data.

For ChIP- or ChIP-Rx-seq, DNA was quantified using the Quant-iT PicoGreen dsDNA assay (Thermo Fisher Scientific). DNA library preparation was done using the NEBnext Ultra II DNA Library Prep Kit (New England Biolabs) following the manufacturer's instructions. Quality of the library was assessed on the Fragment Analyzer (Agilent) using the NGS Fragment High Sensitivity Analysis Kit (1–6000 bp; Agilent). Finally, libraries were subjected to cluster generation and base calling for 75 cycles on Illumina NextSeq 500 platform.

CUT&RUN followed by sequencing was performed as described in *Meers et al., 2019*. 5×10$^5$ cells were harvested using Accutase (Sigma-Aldrich). Cells were washed in wash buffer (20 mM HEPES pH 7.5, 150 mM NaCl, 0.5 mM Spermidine). After washing, cells were coupled to ConA-coated magnetic beads (Polysciences Europe), permeabilised and subsequently incubated with the primary antibody (EXOSC5: NBP2-14952, 1:100, Novus Biologicals) in antibody binding buffer (wash buffer, 0.05% digitonin, 2 mM EDTA) overnight at 4°C. Cells were washed in Dig-Wash buffer (Wash buffer, 0.05% digitonin) and MNase (700 ng/ml) was added to each sample for 1 hr at 4°C. After washing with Dig-Wash buffer and Low-Salt Rinse buffer (20 mM HEPES pH 7.5, 0.5 mM Spermidine, 0.05% digitonin) incubation buffer (3.5 mM HEPES pH 7.5, 10 mM CaCl$_2$, 0.05% digitonin) was added for 30 min at 4°C. STOP buffer (170 mM NaCl, 20 mM EGTA, 0.05% digitonin, 20 µg/ml RNase A, 25 µg/ml glycogen) was added to stop the MNase digestion. DNA fragments were released at 37°C for 30 min. De-crosslinking was performed for 1 hr at 50°C after adding 0.01% SDS and 10 mg/ml proteinase K followed by phenol-chloroform extraction. Precipitated DNA was resuspended in 30 µl 0.1× TE buffer. For library preparation NEBnext Ultra II DNA Library Prep Kit (New England Biolabs) was used. Pre-PCR samples were purified using AmpureXP beads with a ratio of 1.75×. Then the eluted material was used with 16 PCR cycles. The libraries were cleaned using Agentcourt AMPure XP Beads (ratio 0.8×, Beckman Coulter), quality, quantity, and fragment size assessed on the Fragment Analyzer (Agilent) using the NGS Fragment High Sensitivity Analysis Kit (1–6000 bp; Agilent). Finally, libraries were subjected to cluster generation and base calling for 75 cycles on Illumina NextSeq 500 platform.

Phosphorylated linker Hi-C (pLHi-C) was based on Hi-C (*Dixon et al., 2012*) and in situ Hi-C (*Rao et al., 2014*). The main differences consisted in replacing the digestion enzyme and the Klenow fill-in step to increase resolution and overall protocol efficiency. 1×10$^6$ cells per condition were fixed with

formaldehyde (1% final concentration) for 10 min at RT. Reaction was stopped by adding 125 mM glycine for 15 min at RT. Cells were harvested with ice-cold PBS supplemented with protease and phosphatase inhibitors (Sigma-Aldrich). Nuclei isolation was performed by adding buffer I (10 mM Tris-HCl pH 8, 10 mM NaCl, 0.5% NP-40, 1× protease inhibitors; Sigma-Aldrich) coupled with Douncer homogenisation. The resulting solution was centrifuged (2500 × $g$, 5 min, 4°C), the nuclei pellet was washed with 1× NEBuffer 3.1 (NEB) and incubated in buffer IIa (0.2% SDS, 1× NEBuffer 3.1; NEB) for 10 min at 50°C. Buffer IIb (2% Triton X-100, 1.5× NEBuffer 3.1, 1× protease inhibitors) was added to a final volume of 943 µl. Sample was incubated for 15 min at 37°C. 850 U DpnII (NEB) and 35 µl rSAP (NEB) were added and sample was incubated for at least 8 hr at 37°C. Modified DNA oligos, 5'-GATCCCCAAATCT-3' and 5'-GATCAGAT[BtndT]TGGG-3' with 5' end phosphate (Sigma-Aldrich), were annealed (81 µl of each Oligo 100 µM, 1× T4 ligase buffer; NEB) at 98°C for 5 min, followed by 20 min at RT. After digestion, sample was washed with 1× T4 ligase buffer (NEB) (2500 × $g$, 5 min, 4°C). Nuclei pellet was resuspended in buffer IIIa (0.3% SDS, 1× T4 ligase buffer) and incubated for 15 min at 60°C. Buffer IIIb (3% Triton X-100, 1× T4 ligase buffer, 1× protease inhibitors; NEB, Sigma-Aldrich) was added to a final volume of 1 ml and sample was incubated for 15 min at 37°C. The nuclei pellet was resuspended in buffer IV (160 µl annealed oligos, 1.2× T4 ligase buffer, 6000 U T4 ligase, 1× protease inhibitors; NEB) to a final volume of 800 µl after washing it with 1× T4 ligase buffer (NEB) (2500 × $g$, 5 min, 4°C). Sample was incubated overnight at 16°C. rSAP in the digestion followed by phosphorylated linker in ligation was introduced to decrease the likelihood of self-ligation, increasing the final yield. Nuclei pellet was isolated by centrifugation (2500 × $g$, 5 min, 4°C) and mixed with buffer V (10 mM Tris-HCl pH 7.6, 1 mM EDTA, 0.1% SDS, 1× protease inhibitors). Chromatin fragmentation was performed using the Covaris Focused Ultrasonicator M220 for 30 min/ml lysate. Samples were de-crosslinked overnight. Protein and RNA were digested with proteinase K and RNase A, respectively. DNA was isolated by ChIP DNA Clean & Concentrator (Zymo Research) according to the manufacturer's instructions. Biotin pull-down was performed with MyOne Streptavidin C1 beads (Thermo Fisher Scientific) according to the manufacturer's instructions. Library preparation was performed on beads using NEBNext ChIP-Sequencing Library Prep Master Mix Set for Illumina (NEB). The manufacturer's instructions were followed, apart from introducing a washing step between each module of the protocol with buffer VI (3×, 5 mM Tris-HCl pH 8, 0.5 mM EDTA, 1 M NaCl, 0.05% Tween) followed by one with 1× TE buffer. Quality of the library was assessed on the Fragment Analyzer (Agilent) using the NGS Fragment High Sensitivity Analysis Kit (1–6000 bp; Agilent). Libraries were subjected to cluster generation and base calling for 150 cycles paired end on Illumina NextSeq 500 platform.

spLHiChIP and pLHiChIP were based on pLHi-C, ChIP-Rx, and the original HiChIP protocol (*Mumbach et al., 2016*). spLHiChIP and pLHiChIP benefit from the same modules of pLHi-C fixation, digestion, ligation, chromatin fragmentation, and library preparation. The difference consists in the ChIP module, which follows the exact same steps from IP of target protein to de-crosslinked overnight stated in the ChIP workflow. The ChIP module allocated between the chromatin fragmentation and library preparation in pLHi-C workflow defines the spLHiChIP and pLHiChIP workflows. spLHiChIP differs from pLHiChIP by adding murine NHO2A cells as an exogenous control (spike-in) at a 1:20 cell ratio during nuclei isolation. This allows normalisation by murine paired end valid tags in a similar fashion as murine reads for ChIP-Rx. $10 \times 10^6$ cells per condition were fixed with formaldehyde (1% final concentration) for 10 min at RT. Reaction was stopped by adding 125 mM glycine for 15 min at RT. Cells were harvested with ice-cold PBS supplemented with protease and phosphatase inhibitors (Sigma-Aldrich). Nuclei isolation was performed by adding buffer I (10 mM Tris-HCl pH 8, 10 mM NaCl, 0.5% NP-40, 1× protease inhibitors; Sigma-Aldrich) coupled with Douncer homogenisation. The resulting solution was centrifuged (2500 × $g$, 5 min, 4°C), the nuclei pellet was washed with 1× NEBuffer 3.1 (NEB) and incubated in buffer IIa (0.2% SDS, 1× NEBuffer 3.1; NEB) for 10 min at 50°C. Buffer IIb (2% Triton X-100, 1.5× NEBuffer 3.1, 1× protease inhibitors;) was added to a final volume of 943 µl. Sample was incubated for 15 min at 37°C. 850 U DpnII (NEB) and 35 µl rSAP (NEB) were added and sample was incubated for at least 8 hr at 37°C. Modified DNA oligos, 5'-GATCCCCAAATCT-3' and 5'-GATCAGAT[BtndT]TGGG-3' with 5' end phosphate (Sigma-Aldrich), were annealed (81 µl of each Oligo 100 µM, 1× T4 ligase buffer; NEB) at 98°C for 5 min, followed by 20 min at RT. After digestion, sample was washed with 1× T4 ligase buffer (NEB) (2500 × $g$, 5 min, 4°C). Nuclei pellet was resuspended in buffer IIIa (0.3% SDS, 1× T4 ligase buffer) and incubated for 15 min at 60°C. Buffer IIIb (3% Triton X-100, 1× T4 ligase buffer, 1× protease inhibitors; NEB, Sigma-Aldrich) was added to a

final volume of 1 ml and sample was incubated for 15 min at 37°C. The nuclei pellet was resuspended in buffer IV (160 µl annealed oligos, 1.2× T4 ligase buffer, 6000 U T4 ligase, 1× protease inhibitors; NEB) to a final volume of 800 µl after washing it with 1× T4 ligase buffer (NEB) (2500 × $g$, 5 min, 4°C). Sample was incubated overnight at 16°C. Nuclei pellet was isolated by centrifugation (2500 × $g$, 5 min, 4°C) and mixed with buffer V (10 mM Tris-HCl pH 7.6, 1 mM EDTA, 0.1% SDS, 1× protease inhibitors). Chromatin fragmentation was performed using the Covaris Focused Ultrasonicator M220 for 30 min/ml lysate. For each IP reaction, 100 µl Dynabeads Protein A and Protein G (Thermo Fisher Scientific) were pre-incubated overnight with rotation in the presence of 5 mg/ml BSA and 10 µg antibody (MYCN [B8.4.B]: sc-53993, RRID:AB_831602; TFIIIC5: A301-242A, RRID:AB_890669). Chromatin was added to the beads, and IP was performed for a minimum of 6 hr at 4°C with rotation. Beads were washed three times each with washing buffer I (20 mM Tris pH 8.1, 150 mM NaCl, 2 mM EDTA, 1% Triton X-100, 0.1% SDS), washing buffer II (20 mM Tris pH 8.1, 500 mM NaCl, 2 mM EDTA, 1% Triton X-100, 0.1% SDS), washing buffer III (10 mM Tris pH 8.1, 250 mM LiCl, 1 mM EDTA, 1% NP-40, 1% sodium deoxycholate, including a 5 min incubation with rotation) and TE buffer. Chromatin was eluted twice by incubating with 200 µl elution buffer (100 mM NaHCO$_3$, 1% SDS in TE) for 15 min with rotation. Input samples and eluted samples were de-crosslinked overnight. Protein and RNA were digested with proteinase K and RNase A, respectively. DNA was isolated by phenol-chloroform extraction followed by an ethanol precipitation. DNA library preparation was done using the NEBnext Ultra II DNA Library Prep Kit (New England Biolabs) following the manufacturer's instructions. Quality of the library was assessed on the Fragment Analyzer (Agilent) using the NGS Fragment High Sensitivity Analysis Kit (1–6000 bp; Agilent). Finally, libraries were subjected to cluster generation and base calling on Illumina NextSeq 500 platform.

RNA-seq was performed by extracting RNA with RNeasy Mini Kit (QIAGEN) according to the manufacturer's instructions. On-column DNase I digestion was performed followed by mRNA isolation with the NEBNext Poly (A) mRNA Magnetic Isolation Kit (NEB). Library preparation was done with the Ultra II Directional RNA Library Prep for Illumina following the manufacturer's manual. Libraries were size selected using SPRIselect Beads (Beckman Coulter) after amplification with nine PCR cycles. Library quantification and size determination was performed with the Fragment Analyzer (Agilent) using the NGS Fragment High Sensitivity Analysis Kit (1–6000 bp; Agilent). Libraries were subjected to cluster generation and base calling for 100 cycles paired end on Illumina NextSeq 2000 platform.

## Bioinformatics analysis and statistics

All libraries were subjected to NextSeq 500 or NextSeq 2000 (Illumina) sequencing following the manufacturer's guidelines. Base call data was converted and demultiplex to FASTQ files by bcl2fastq Conversion Software v1.1.0 (Illumina). Quality control of each dataset was done using FastQC tool.

ChIP-seq and CUT&RUN sequencing reads were aligned to human (hg19/GRCh37 assembly) using Bowtie2 v2.3.5.1. with default parameters (*Langmead and Salzberg, 2012*). Normalisation by aligned read number was performed on the input and on all correspondent IP samples.

For each sample, the number of spike-in normalised reads was calculated by dividing the number of reads mapping exclusively to hg19 by the number of reads mapping exclusively to mm10 and multiplying this number with the smallest number of reads mapping to mm10 of all samples. After normalisation, files were converted to bedGraph with 'bedtools genomecov' v2.26 (*Quinlan and Hall, 2010*). These files were used to plot genome browser track examples via the package plotgardener v1.012 in R v4.1.1.

MACS2 v2.1.2 (*Zhang et al., 2008*) was used for peak calling with the following ChIP-seq datasets and parameters, with inputs serving as control: MYCN (GSM3044606, GSM3044608; dup = 5, q=1 × 10$^{-3}$), TFIIIC5 in SH-EP-MYCN-ER (this work; dup = 5, q=10$^{-3}$), and RAD21 (this work; dup = 5, p=10$^{-2}$). H3K4me1, H3K4me3, and H3K27ac peaks were called with SICER v1.1 (*Xu et al., 2014*), using input as control and the parameters: redundancy threshold = 1, window size = 200 bp, fragment size = 75 bp, effective genome fraction = 0.74, gap size = 600, FDR = 0.001. Peak calling results were refined and confirmed by visual inspection.

RNA-seq reads were aligned to human (hg19/GRCh37 assembly) using STAR aligner v2.7.9a with the default parameters and gene quantification (--quantMode GeneCounts) (*Dobin et al., 2013*). The output table with the gene counts was loaded into R v4.1.1. and processed with DESeq2 package accordingly to their manual (*Love et al., 2014*). Genes with or more than 15 counts in at

least three samples or more were selected, and all samples were normalised by size factors. The fold average for three independent biological replicates was calculated on treatment to control as shown in the correspondent figure and legend. The result was averaged in 100 or 150 bins for a total of 14,085 genes and displayed as dot plot. Alternative splicing events were identified using rMATS (*Shen et al., 2014a*).

pLHi-C, pLHiChIP, and spLHiChIP sequencing reads were processed in two steps. In the first step they were all processed in the same way, HiC-Pro v 2.11.4 (*Servant et al., 2015*) trims the reads for the linker sequence, aligns it to hg19, and filters it by DpnII restriction fragments of the same genome assembly. The default parameters were used for the alignment steps, except for the length of the seed (-L) parameter that was set to 25 for the global options and 15 for the local one, and mismatch (-N) that was set to 1. Read trimming was performed by inserting combinations of the linker sequence (GATCAGATTTGGGGATC, GATCCCCAAATCTGATC) in the ligation site parameter. Reads considered as duplicates were accepted and multi-mapped reads were only removed for pLHiChIP and spLHiChIP. Quality control for all steps until all valid pairs (paired end tags [PETs]) output was also performed through HiC-Pro. We compared HiC-Pro's standard quality controls example (IMR90 replicate 1 sample; GSM862724), according to *Servant et al., 2015*, to the base of our methods variation, pLHi-C.

In the second step, spLHiChIP samples undergo the same workflow described for the first step replacing hg19 for mm10 assembly. PETs from mm10 were removed from hg19 PETs and a spike-in normalisation was applied to hg19 PETs in the same way explained for ChIP-Rx hg19 aligned reads. DNA loop calling was performed on spLHiChIP and pLHiChIP PETs using hichipper v0.7.7 (*Lareau and Aryee, 2018*). ChIP-seq peak calling sites were used as pre-determined peaks. All the default parameters with a loop maximum distance (--max-dist) of 5,000,000 were used for all proteins. We compared the DNA loops calling quality control for Oct4 HiChIP (GSM2238510) from the original method description (*Mumbach et al., 2016*) to our method variation for MYCN and TFIIIC5. An extra quality control to show enrichment and specificity was performed by plotting MYCN pLHi-ChIP, its input pLHi-C, and its ChIP-seq with pLHiChIP and pLHi-C normalised to the same number of PETs. DNA loops calling results were refined and confirmed by visual inspection. The mango output files from the loop calling were then loaded in R, filtered by a q-value equal or smaller than 0.01, and used in all further plots and analyses. All examples showing DNA loops, ChIP-seq, and annotation tracks in different genomic regions/locus were plotted using plotgardener package in R environment.

The different genome functional annotations were retrieved from UCSC table browser (*Karolchik et al., 2004*). Promoters, gene bodies, and TES come from Reference Sequence (RefSeq) collection of genes for hg19 assembly and were adapted via R language. For RNAPII targets, promoters were defined as TSS ± 0.5 kb (*Büchel et al., 2017*), gene bodies as the sum of exons and introns, TES as TES ± 0.3 kb. Enhancers were defined as in *Walz et al., 2014*: H3K3me1 and H3K27ac overlapping sites without H3K4me3. Overlapping and discontinuous regions were obtained by using peak calling outputs as inputs in intersectBed (BEDTools) with standard parameters.

Overlapping between different pLHiChIP was calculated by 'bedtools pairtopair' function (BEDTools) using the options to prevent self-hits and ignore strands. The type of overlap was set to report when both anchors of the different pLHiChIP overlap (-type either). This provided outputs of loops with and without overlapping that could be used for other analyses. For discrimination between the different types of overlaps, the type both was removed from the type either and the two types were then reported. Overlaps were called joint sites and adapted as Venn diagrams by centring the coordinates and numbers to MYCN anchors or MYCN-TFIIIC5 anchors. TADs overlap corresponds to common sites in at least eight out of nine cell lines reported in (GSE35156) (*Rao et al., 2014*).

The GenomicInteractions v1.28 (*Harmston et al., 2015*) package in R language was used to calculate and report the overlaps between the different pLHiChIP anchors and their functional annotations. Overlaps on different functional annotations were reported in the following order of priority: enhancer, promoter, TES, gene body, tRNA, rRNA, miRNA, snRNA, snoRNA, SINE, LINE, LTR, and satellite. The bar graphs reporting the functional annotation of each anchor are proportional to the

total number of loops of each pLHiChIP, numbers were rounded and plotted by GenomicInteractions and ggplot2 v3.3.5 packages in R environment.

Motif analysis was performed by using the FIMO algorithm (*Grant et al., 2011*) of the MEME Suite software toolkit v5.3.3 (*Bailey et al., 2015*). The function getfasta (BEDTools) extracted the hg19 FASTA sequence from each pLHiChIP anchor to be used independently as input for motif search. The background model was generated by the function fasta-get-markov (MEME Suite) using hg19 FASTA with standard parameters. Pre-defined motifs for MYCN (E-box) (MA0104.4: JASPAR 2018 *Sandelin et al., 2004*), B-box, and A-box were scanned using the options '`--thresh 1.0 --text`' with the default parameters (*Chan et al., 2021*; *Pavesi et al., 1994*). Resulting hits were then filtered by p-value $\leq 1 \times 10^{-4}$. In R environment, duplicated hits of the same motif in the same anchor were filtered by the one with the lowest p-value. Different motifs in the same anchor window were reported according to the decreasing priority: E-box, B-box, and A-box. The overlap between anchors and motifs was computed by GenomicInteractions package and the corresponding heatmap was plotted in R.

To visualise the influence of the MYCN interaction on gene expression (*Figure 3C*), MYCN-bound genes (as defined by MACS2 on MYCN ChIP-seq data, above) were subdivided into genes that are bound by MYCN without loops ('no MYCN loops') and genes bound by MYCN forming loops (above, 'MYCN loops'). For both classes, RNA-seq reads between TSS and TSS+2 kb were counted using 'bedtools intersect' on spike-normalised ChIP-seq data for RNAPII (A10), MYCN (GSM3044606), TFIIIC5; processed transcript data were obtained from *Büchel et al., 2017* (RNA-seq, GSE111905; *Büchel et al., 2017*) and *Papadopoulos et al., 2022* (4sU-sequencing, GSE164569; *Papadopoulos et al., 2022*), respectively. Plots were generated with 'boxplot' and statistical significance calculated using Student's t-test (unpaired, two-sided, unequal variance), both in R v3.6.3.

Over-representation analysis for Molecular Signatures Database (MsigDB *Subramanian et al., 2005*) were calculated by clusterProfiler package (*Wu et al., 2021*) in R environment. Different functional classes of loops were retrieved from the overlaps and discontinuous loops by the GenomicInteractions package. Gene names were converted to HGNC symbol name using AnnotationDbi package and sorted by the number of PETs. Duplicated gene names were merged and the median of their number of PETs was used. A hypergeometric test was calculated on the sorted list for MSigDB collections H (hallmark gene sets) or C5 (ontology gene sets) using a 0.05 q-value threshold.

MYCN spLHiChIP-treated sample was normalised by its control counterpart in each biological replicate before plotted as graph bar in R environment. The ratio of TFIIIC5 binding (ChIP-Rx) per number of interaction (spLHiChIP) was calculated in R environment v4.1.1. The number of ChIP-Rx reads was calculated to TSS ± 2 kb for all genes using bedtools intersect (BEDtools) with default parameters. The number of spLHiChIP interactions was calculated for the same region. The ratio was calculated on reads to number of interactions per each sample and plotted as bar graph.

Networks were reconstituted by using the R package igraph v1.2.11 and Cytoscape v3.9 (*Brockmann et al., 2013*). The different classes of loops were retrieved via the GenomicInteractions package and converted to the simple interaction file format and to a node table. These files were used in Cytoscape for visualisation, using the Edge-weighted Spring-Embedded Layout based on the number of PETs for each loop. The resulting MYCN network reports the nodes as functional annotations, as explained before. In parallel, the sif file was used as input for the igraph package. This package was used to define the membership of each anchor to the different clusters and to quantify the number of nodes per cluster. The statistical significance comparing the number of nodes per cluster between two groups were calculated via Wilcoxon rank sum test.

Density plots for the different regions were performed using ngs.plot v2.41.3 (*Shen et al., 2014b*). Default parameters were used for samples normalised by reads number. For spike-in normalised samples, we used a custom variation of ngs.plot that allows the spike-in normalisation by skipping the internal normalisation step. All densities plots and metaplots were based on gene expression for SH-EP-MYCN-ER cells ± MYCN induction (4-OHT treatment) – evaluated by RNA-seq (GSE111905; *Herold et al., 2019*). Based on this, all expressed gene sets consist of 14,704 genes, down- and up-regulated gene sets are represented by 613 and 921 genes, respectively. MYCN down- and

up-regulated genes are defined by negative and positive fold change upon ±4-OHT treatment of genes with q-values less than 0.05 and expression filter by counts per million. In R v4.1.1, Wilcoxon rank sum test (unpaired, two-sided) was used to compare the different conditions on the plotted region per ChIP.

PLA's p-values were calculated by comparing the signal of the pool of the cells in all replicates using Wilcoxon rank sum test (unpaired, two-sided) in R v4.1.1.

## Acknowledgements

We thank Vanessa Luzak and Yordan Sbirkov for initial experiments on chromatin architecture. This work was funded by grants from German Cancer Aid (Mildred Scheel Early Career Center, #70113303; GB), German Cancer Aid (#70113870; ME), German Research Foundation (DFG) (INST 93/1023–1-FUGG and EI222/21-1; ME), Alex's Lemonade Stand Foundation (Crazy 8 Initiative; ME and GB), Wilhelm Sander-Stiftung (#2023.002.1; SH) and Medical Research Council (MR/V029975/1; EL and RB).

## Additional information

### Competing interests

Martin Eilers: founder and shareholder of Tucana Biosciences. The other authors declare that no competing interests exist.

### Funding

| Funder | Grant reference number | Author |
|---|---|---|
| Deutsche Krebshilfe | 70113303 | Gabriele Büchel |
| Deutsche Krebshilfe | 70113870 | Martin Eilers |
| Deutsche Forschungsgemeinschaft | INST 93/1023-1-FUGG and EI222/21-1 | Martin Eilers |
| Alex's Lemonade Stand Foundation for Childhood Cancer | Crazy 8 Initiative | Martin Eilers Gabriele Büchel |
| Wilhelm Sander-Stiftung | 2023.002.1 | Steffi Herold |
| Medical Research Council | MR/V029975/1 | Eoin Leen Richard Bayliss |

The funders had no role in study design, data collection and interpretation, or the decision to submit the work for publication.

### Author contributions

Raphael Vidal, Conceptualization, Formal analysis, Investigation, Writing – review and editing; Eoin Leen, Steffi Herold, Funding acquisition, Investigation; Mareike Müller, Daniel Fleischhauer, Christina Schülein-Völk, Dimitrios Papadopoulos, Isabelle Röschert, Investigation; Leonie Uhl, Carsten P Ade, Peter Gallant, Formal analysis; Richard Bayliss, Supervision, Funding acquisition; Martin Eilers, Conceptualization, Supervision, Funding acquisition, Writing – original draft, Writing – review and editing; Gabriele Büchel, Conceptualization, Supervision, Funding acquisition, Investigation, Visualization, Writing – original draft, Writing – review and editing

### Author ORCIDs

Dimitrios Papadopoulos https://orcid.org/0000-0002-3415-7407
Carsten P Ade https://orcid.org/0000-0001-7226-1179
Richard Bayliss https://orcid.org/0000-0003-0604-2773
Martin Eilers https://orcid.org/0000-0002-0376-6533
Gabriele Büchel https://orcid.org/0000-0001-7070-7341

Reviewer #1 (Public review): https://doi.org/10.7554/eLife.94407.3.sa1
Reviewer #2 (Public review): https://doi.org/10.7554/eLife.94407.3.sa2
Reviewer #3 (Public review): https://doi.org/10.7554/eLife.94407.3.sa3
Author response https://doi.org/10.7554/eLife.94407.3.sa4

## Additional files

### Supplementary files
- MDAR checklist
- Source code 1. Analysis of ChIP sequencing data.
- Source code 2. Analysis of HiC and HiChIP data.

### Data availability

The following published datasets were used: MYCN ChIP-seq and RNA-seq: GSE111905; 4sU-seq: GSE164569. Data from this work can be found under GEO accession GSE223058. All data generated or analysed during this study are included in the manuscript and supporting files; source data files have been provided. The analysis of the sequencing data is described in the methods and the source code is included in the source code files.

The following dataset was generated:

| Author(s) | Year | Dataset title | Dataset URL | Database and Identifier |
|---|---|---|---|---|
| Vidal R, Leen E, Müller M, Schülein-Völk C, Eilers U, Papadopoulos D, Röschert I, Herold S, Ade CP, Gallant P, Bayliss R, Eilers M, Büchel G | 2024 | TFIIIC and MYCN link the three-dimensional chromatin structure of promoters to transcription termination of stalled RNA polymerase | https://www.ncbi.nlm.nih.gov/geo/query/acc.cgi?acc=GSE223058 | NCBI Gene Expression Omnibus, GSE223058 |

The following previously published datasets were used:

| Author(s) | Year | Dataset title | Dataset URL | Database and Identifier |
|---|---|---|---|---|
| Herold S, Kalb J, Büchel G, Ade CP | 2019 | Recruitment of BRCA1 limits MYCN-driven accumulation of stalled RNA Polymerase in neuroblastoma | https://www.ncbi.nlm.nih.gov/geo/query/acc.cgi?acc=GSE111905 | NCBI Gene Expression Omnibus, GSE111905 |
| Papadopoulos D, Solvie D, Baluapuri A, Endres T | 2021 | The MYCN oncoprotein resolves conflicts of stalling RNA Polymerase with the replication fork | https://www.ncbi.nlm.nih.gov/geo/query/acc.cgi?acc=GSE164569 | NCBI Gene Expression Omnibus, GSE164569 |

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

# Appendix 1

## Appendix 1—key resources table

| Reagent type (species) or resource | Designation | Source or reference | Identifiers | Additional information |
|---|---|---|---|---|
| Cell line (human) | SH-EP | Schwab | CVCL_RR78 | |
| Cell line (human) | SH-EP-MYCN-ER | Eilers | | https://doi.org/10.1038/s41586-019-1030-9 |
| Cell line (murine) | NIH-3T3 | ATCC | CVCL_0594, Cat# CRL-1658 | |
| Cell line (murine) | NHO2A | Schramm | | https://doi.org/10.1080/2162402X.2015.1131378 |
| Cell line (human) | HEK293TN | ATCC | CVCL_UL49, Cat# CRL-11268 | |
| Cell line (insect) | SF9 | Gibco | Cat# 11496015 | Recombinant protein expression |
| Strain, strain background (*Escherichia coli*) | BL21(DE3)RIL | Merck | Cat# 69,450M | |
| Antibody | TFIIIC90 (rabbit polyclonal) | Bethyl Laboratories | Cat# A301-239A RRID:AB_890667 | WB (1:2000) |
| Antibody | TFIIIC5 (rabbit polyclonal) | Bethyl Laboratories | Cat# A301-242A RRID:AB_890669 | WB (1:1000) Seq (10–15 µg) PLA (1:1000) |
| Antibody | TFIIIC102 (rabbit polyclonal) | Bethyl Laboratories | Cat# A301-238A RRID:AB_890671 | WB (1:2000) |
| Antibody | TFIIIC110 (mouse monoclonal) | Santa Cruz | Cat# sc-81406 RRID:AB_2115237 | WB (1:1000) |
| Antibody | MYCN (B8.4.B) (mouse monoclonal) | Santa Cruz | Cat# sc-53993 RRID:AB_831602 | WB (1:1000) Seq (10–15 µg) ChIP (3 µg) |
| Antibody | TFIIIC35 (rabbit polyclonal) | Novus Biologicals | Cat# NBP2-31851 RRID:AB_2891101 | WB (1:1000) |
| Antibody | TFIIIC1 (rabbit polyclonal) | Novus Biologicals | Cat# NBP2-14077 RRID:AB_2891102 | WB (1:1000) |
| Antibody | FLAG (mouse monoclonal) | Sigma-Aldrich | Cat# F1804 RRID:AB_262044 | WB (1:2000) |
| Antibody | MYC (Y69) (rabbit monoclonal) | abcam | Cat# ab32072 RRID:AB_731658 | WB (1:1000) ChIP (3 µg) |
| Antibody | Vinculin (hVin-1) (mouse monoclonal) | Sigma-Aldrich | Cat# V9131 RRID:AB_477629 | WB (1:5000) |
| Antibody | GAPDH (14C10) (rabbit monoclonal) | Cell Signaling | Cat# 2118 RRID:AB_561053 | WB (1:5000) |
| Antibody | RNAPII (F12) (mouse monoclonal) | Santa Cruz | Cat# sc-55492 RRID:AB_630203 | PLA (1:2000) |
| Antibody | NELFE (rabbit polyclonal) | Merck | Cat# ABE48 RRID:AB_10806770 | PLA (1:1000) |
| Antibody | PP2A (rabbit polyclonal) | Cell Signaling | Cat# 2038 RRID:AB_2169495 | PLA (1:1000) |
| Antibody | BRCA1 (rabbit polyclonal) | Bethyl Laboratories | Cat# A300-000A RRID:AB_67367 | ChIP (3 µg) |
| Antibody | PNUTS (rabbit polyclonal) | Bethyl Laboratories | Cat# A300-439-A RRID:AB_420948 | PLA (1:1000) |
| Antibody | XRN2 (rabbit polyclonal) | Bethyl Laboratories | Cat# A301-103-A RRID:AB_2218876 | PLA (1:2000) |
| Antibody | RNA polymerase II CTD repeat YSPTSPS (phospho Ser2) (rabbit polyclonal) | Abcam | Cat# ab5095 RRID:AB_304749 | Seq (10–15 µg) |
| Antibody | RNA polymerase II (unphosphorylated, 8WG16) (mouse monoclonal) | Santa Cruz | Cat# sc-56767 RRID:AB_785522 | Seq (10–15 µg) |

*Appendix 1 Continued on next page*

*Appendix 1 Continued*

| Reagent type (species) or resource | Designation | Source or reference | Identifiers | Additional information |
|---|---|---|---|---|
| Antibody | EXOSC5 (rabbit polyclonal) | Novus Biologicals | Cat# NBP2-14952 | C&R (1:100) |
| Antibody | Donkey Anti-rabbit HRP (polyclonal secondary) | Amersham | Cat# NA934 RRID:AB_772206 | WB (1:3000) |
| Antibody | Sheep Anti-mouse HRP (monoclonal secondary) | Amersham | Cat# NA931 RRID:AB_772210 | WB (1:3000) |
| Recombinant DNA reagent | pInducer11 | Addgene | Cat# 44363 *Meerbrey et al., 2011* | Inducible lentiviral gene silencing vector |
| Recombinant DNA reagent | LT3GEPIR | Addgene | Cat# 111177 Zuber | Tet-ON miR-E (miR-30 variant)-based RNAi |
| Recombinant DNA reagent | psPAX.2 | Addgene | Cat# 12260 Trono | Second-generation lentiviral packaging plasmid |
| Recombinant DNA reagent | pMD2.G | Addgene | Cat# 12259 Trono | VSV-G envelope expressing plasmid |
| Sequence-based reagent | shRNA targeting TFIIIC5 | *Fellmann et al., 2013* | shRNA ID: GTF3C5.1361 | AAGCGCAGCAC CTACAACTACA |
| Sequence-based reagent | shRNA targeting TFIIIC5 | *Pelossof et al., 2017* | shRNA ID: GTF3C5_9328_847 | TTGATAAATCTTG GCATCTGGG |
| Sequence-based reagent | shRNA targeting TFIIIC2 | *Pelossof et al., 2017* | shRNA ID: GTF3C2_2976_2623 | TGAAGCAGAAG AATGGTCTGGA |
| Sequence-based reagent | shRNA targeting TFIIIC3 | *Policarpi et al., 2017* | shRNA ID: GTF3C3_9330_545 | TTCATCATTTTCTTGGTTTCAC |
| Sequence-based reagent | TFAP4 | This paper | ChIP qPCR Primer | (forward: CCGGGCGC TGTTTACTA; reverse: CAGGACACGGAG AACTACAG) |
| Sequence-based reagent | POLG | This paper | ChIP qPCR Primer | (forward: CTTCTCAAGG AGCAGGTGGA; reverse: TCATAACCTCC CTTCGACCG) |
| Sequence-based reagent | NPM1 | This paper | ChIP qPCR Primer | (forward: TTCACCG GGAAGCATGG; reverse: CACGCGAGG TAAGTCTACG) |
| Sequence-based reagent | Intergenic region | This paper | ChIP qPCR Primer | (forward: TTTTCTCAC ATTGCCCCTGT; reverse: TCAATGCTGTA CCAGGCAAA) |
| Sequence-based reagent | NCL | This paper | ChIP qPCR Primer | (forward: CTACCACCC TCATCTGAATCC; reverse: TTGTCTCGC TGGGAAAGG) |
| Sequence-based reagent | NME1 | This paper | ChIP qPCR Primer | (forward: GGGGTGGAG AGAAGAAAGCA; reverse: TGGGAGTAG GCAGTCATTCT) |
| Sequence-based reagent | PLD6 | This paper | ChIP qPCR Primer | (forward: GCTGTGGGTCCCGGATTA; reverse: CCTCCAGAGTCAGAGCCA) |
| Sequence-based reagent | TAF4B | This paper | ChIP qPCR Primer | (forward: AAGGTCGT CGCTCACAC, reverse: GCGTGGCTATA TAAACATGGCT) |
| Sequence-based reagent | RPL22 | This paper | ChIP qPCR Primer | (forward: CCGTAGCTTC CTCTCTGCTC, reverse: ACCTCTTGGG CTTCCTGTCT) |
| Sequence-based reagent | CCND2 | This paper | ChIP qPCR Primer | (forward: GCCAGCTGC TGTTCTCCTTA, reverse: CCCCTCCTC CTTTCAATCTC) |
| Sequence-based reagent | DNA oligos for Hi-C | This paper | DNA oligos for Hi-C | GATCCCCAAATCT |
| Sequence-based reagent | DNA oligos for Hi-C | This paper | DNA oligos for Hi-C | GATCAGAT[BtndT]TGGG |
| Commercial assay or kit | Duolink In Situ PLA Probe Anti-Rabbit PLUS, Affinity purified Donkey anti-Rabbit IgG (H+L) | Sigma-Aldrich | Cat# DUO92002 | |

*Appendix 1 Continued on next page*

*Appendix 1 Continued*

| Reagent type (species) or resource | Designation | Source or reference | Identifiers | Additional information |
|---|---|---|---|---|
| Commercial assay or kit | Duolink In Situ PLA Probe Anti-Mouse MINUS, Affinity purified Donkey anti-Mouse IgG (H+L) | Sigma-Aldrich | Cat# DUO92004 | |
| Commercial assay or kit | Duolink In Situ Detection Reagents Red | Sigma-Aldrich | Cat# DUO92008 | |
| Commercial assay or kit | Duolink In Situ Wash Buffers, Fluorescence | Sigma-Aldrich | Cat# DUO82049 | |
| Commercial assay or kit | RNeasy Mini Kit (250) | QIAGEN | Cat# 74106 | |
| Commercial assay or kit | RNase-free DNase kit | QIAGEN | Cat# 79254 | |
| Commercial assay or kit | NEBNext Ultra II Directional RNA Second Strand Module | NEB | Cat# E7550 L | |
| Commercial assay or kit | NEBNext Poly(A) mRNA Magnetic Isolation Module | NEB | Cat# E7490 L | |
| Commercial assay or kit | NEBNext ChIP-Seq Library Prep Master Mix Set for Illumina | NEB | Cat# E6240 L | |
| Commercial assay or kit | NEBNext Ultra II DNA Library Prep Kit for Illumina | NEB | Cat# E7645 L | |
| Commercial assay or kit | NEBNext Multiplex Oligos for Illumina (Dual Index Primers Set 1) | NEB | Cat# E7600 S | |
| Commercial assay or kit | NextSeq 500/550 High Output Kit v2 (75 cycles) | Illumina | Cat# FC-404-2005 | |
| Commercial assay or kit | NextSeq 1000/2000 P2 Reagents (100 Cycles) v3 | Illumina | Cat# 20046811 | |
| Commercial assay or kit | Quant-iT Pico Green | Thermo Fisher Scientific Inc | Cat# P7589 | |
| Commercial assay or kit | NGS Fragment High Sensitivity Analysis Kit (1–6000 bp) | Agilent | Cat# DNF-474-0500 | |
| Commercial assay or kit | NGS Fragment High Sensitivity Small DNA Fragment Analysis Kit, 50–1500 bp | Agilent | Cat# DNF-477-0500 | |
| Commercial assay or kit | Standard Sensitivity RNA Analysis Kit (15 nt), 500 samples | Agilent | Cat# DNF-471-0500 | |
| Commercial assay or kit | ChIP DNA Clean & Concentrator | Zymo Research Europe GmbH | Cat# D5205 | |
| Chemical compound, drug | DRB | Sigma-Aldrich | Cat# D1916-50MG | |
| Chemical compound, drug | Doxycycline | Sigma-Aldrich | Cat # D9891-1G | |
| Chemical compound, drug | Polybrene | Sigma-Aldrich | Cat# 107689-100G | |
| Chemical compound, drug | 4-Hydroxytamoxifen | Sigma-Aldrich | Cat# H7904-5MG | |
| Chemical compound, drug | X-tremeGENE HP Transfection Reagent | Roche | Cat# 06 366 244 001 | |
| Chemical compound, drug | Hoechst 33342 | Sigma-Aldrich | Cat# B2261 | |
| Chemical compound, drug | Dynabeads Protein A | Life Technologies GmbH | Cat# 10002D | |
| Chemical compound, drug | Dynabeads Protein G | Life Technologies GmbH | Cat# 10004D | |
| Chemical compound, drug | Formaldehyde (37%) | Roth | Cat# 4979.1 | |
| Chemical compound, drug | ConA-coated magnetic beads | Polysciences Europe | Cat# 86057-10 | |

*Appendix 1 Continued on next page*

*Appendix 1 Continued*

| Reagent type (species) or resource | Designation | Source or reference | Identifiers | Additional information |
|---|---|---|---|---|
| Chemical compound, drug | AmpureXP beads (SPRI select reagent) | Beckman Coulter | Cat# B23318 | |
| Chemical compound, drug | MyOne Streptavidin C1 beads | Thermo Fisher Scientific | Cat# 65601 | |
| Chemical compound, drug | Accutase | Sigma-Aldrich | Cat# A6964-500ML | |
| Chemical compound, drug | Digitonin | Merck | Cat# 300410-1GM | |
| Chemical compound, drug | DpnII | NEB | Cat# R0543M | |
| Chemical compound, drug | rSAP | NEB | Cat# M0371L | |
| Chemical compound, drug | T4 DNA Ligase | NEB | Cat# M0202M | |
| Software, algorithm | Bcl2fastq Conversion Software v1.1.0 | Illumina | | |
| Software, algorithm | FastQC v0.11.5 | *Wingett and Andrews, 2018* | | |
| Software, algorithm | Bowtie2 v2.3.5.1 | *Langmead and Salzberg, 2012* | | |
| Software, algorithm | Bedtools v2.26 | *Quinlan and Hall, 2010* | | |
| Software, algorithm | plotgardener v1.012 | *Kramer et al., 2022* | | |
| Software, algorithm | Integrated Genome Browser v9.1.6 | *Nicol et al., 2009* | | |
| Software, algorithm | R v4.1.1 and v.3.6.3 | *R Development Core Team, 2022* | | |
| Software, algorithm | MACS v2.1.2 | *Zhang et al., 2008* | | |
| Software, algorithm | SICER v1.1 | *Xu et al., 2014* | | |
| Software, algorithm | STARaligner v2.7.9a | *Dobin et al., 2013* | | |
| Software, algorithm | DESeq2 v1.34 | *Love et al., 2014* | | |
| Software, algorithm | HiC-Pro v2.11.4 | *Servant et al., 2015* | | |
| Software, algorithm | hichipper v0.7.7 | *Lareau and Aryee, 2018* | | |
| Software, algorithm | GenomicInteractions v1.28 | *Harmston et al., 2015* | | |
| Software, algorithm | ggplot2 v3.3.5 | *Wickham, 2009* | | |
| Software, algorithm | MEME Suite software toolkit v5.3.3 | *Bailey et al., 2015* | | |
| Software, algorithm | clusterProfiler v4.2.2 | *Wu et al., 2021* | | |
| Software, algorithm | AnnotationDbi v1.56.2 | *Pagès et al., 2024* | | |
| Software, algorithm | igraph v1.2.11 | *Csardi and Nepusz, 2006* | | |
| Software, algorithm | Cytoscape v3.9 | *Shannon et al., 2003* | | |
| Software, algorithm | GSEA v4.0.2 | *Subramanian et al., 2005* | | |
| Software, algorithm | ngs.plot v2.41.3 | *Shen et al., 2014b* | | |
| Software, algorithm | biomaRt v 2.40.5 | *Durinck et al., 2005* | | |
| Software, algorithm | Prism 5.0 Software | GraphPad | | |
| Software, algorithm | Operetta CLS High Content Imaging System | PerkinElmer | | |
| Software, algorithm | Harmony High Content Imaging and Analysis Software | PerkinElmer | | |

