## [Editor Report · eLife assessment]

This study presents the **valuable** finding that TFIIIC interacts with MYCN to regulate RNA polymerase II dynamics by dissecting its impact on 3D chromatin architecture. Authors provide **convincing** evidence that MYCN and TFIIIC show long-range chromatin contacts, and that the expression of each protein limits the function of the other. The notion emerges that TFIIIC helps MYCN to maintain output at promoters while decreasing less productive associations at larger more extensively connected chromatin hubs. The paper is of interest to molecular biologists working on MYCN-dependent regulation of gene expression.

---

## [Referee Report · Reviewer #1 (Public review)]

Summary:

In this manuscript entitled "Association with TFIIIC limits MYCN accumulation in hubs of active promoters and chromatin accumulation of non-phosphorylated RNA polymerase II" the authors examine how the cohesin complex component (and RNA pol III associated factor) TFIIIC interacts with MYCN and controls transcription. They confirm that TFIIIC co-purifies with MYCN, dependent on its amino terminus, as shown in previous work. The authors also find that TFIIIC and MYCN are both found in promoter hubs and suggest that TFIIIC inhibits MYCN association with these hubs. Finally, the authors indicate that TFIIIC/MYCN alter exosome function, and BRCA1 dependent effects, at MYCN regulated loci.

In the revised manuscript the authors have adequately addressed or responded to our questions and comments. The exception concerns point #2 in our initial review:

(2) The authors indicate in Figure 2 that TF3C has essentially no effect on MYCN- dependent gene expression and/or transcription elongation. Yet a previous study (PMID: 29262328) associated with several of the same authors concluded that TF3C positively affects transcription elongation. The authors to not attempt to reconcile these disparate results and the point still needs to clarified.

Authors' Response

We agree that the data in this manuscript do not support the role on transcription elongation. This point was also raised by Reviewer 3. Comparing our new results to the data published previously we can summarize that the data sets in the two studies show three key results: First, the traveling ratio of RNAPII changes upon induction of MYCN. Second, RNAPII decreases at the transcription start side and third, it increases towards the end side.

We agree that in the previous study we linked the traveling ratio directly to elongation. However performing ChIP-seq with different RNAPII antibodies showed us that for example RNAPII (N20), which is unfortunately discontinued, gives different results compared to RNAPII (A10). Combining our new results using the RNAPII (8WG16) antibody shows that the traveling ratio is not only reflecting transcription elongation but also includes that the RNAPII is kicked-off chromatin at the start side.

Reviewer revised response:

The explanation for the change in interpretation of the previous study (Buchel, et. al., 2017) in light of the differing results using different RNA pol2 antibodies used in the present study seems reasonable. However the final manuscript may well result in some confusion in the literature in regards to TF3C and elongation. This is because, while the authors refer to the earlier paper frequently, they do not directly discuss the re-interpretation of the elongation conclusion of the earlier paper. It seems likely that a reader of the present paper will find this issue confusing when trying to reconcile the results of the two papers.

---

## [Referee Report · Reviewer #2 (Public review)]

This manuscript reports several interesting observations that invite follow-up. The notion that hubs, and perhaps condensates that may (or may not embrace them) are functionally and physiologically important is an open issue at this time. The authors note that TFIIIC helps to prune extraneous connections from hubs, but do not comment that the connections that are maintained are also reinforced. At the same time only modest changes in gene expression associated with expanded or decreased connections and changes in bound proteins. One interesting possibility might be that standard methods for assessing expression miss changes global or background transcription. It seems that the TFIIIC-MYCN-ER connection has features that would help to suppress such background. The results invite a more global consideration of TFIIIC than as primarily RNAPIII/small RNA transcription factor and of MYCN as an E-box dependent transcription factor. The results use sate of the art methods to develop interesting new ideas that have the potential to instruct further studies that may reveal new mechanisms of action for TFIIIC and MYCN.

The work is however subject to a couple of caveats. First, the authors should be more cautious when drawing firm conclusions about the dynamics and kinetics of transcription from the static snapshots obtained from most genomic methods. For example, please take a look at Figure 1F of "Transcription elongation defects link oncogenicSF3B1 mutations to targetable alterations in chromatin landscape" by Buddu et al, https://doi.org/10.1016/j.molcel.2024.02.032. Here, an increase in RNAPSer2P is seen in gene bodies and a bit at the TES- superficially inviting the conclusion that expression is increased (a similar erroneous conclusion has been claimed in other genomic studies), but the increase is in fact, not due to increased transcription, rather to impaired elongation-this conclusion required performing TT-Seq which allowed inferences to be made about elongation rates. Acknowledging this qualification would help advise the reader.

The authors also need to discuss directly what differences between the MYC predominant SH-EP cells and the MYCN-predominant SH-EP-MYCNER+tamoxifen are qualitative versus quantitative. MYCNER indeed associates much more with chromatin than did MYC, but there seems to be a lot more MYCER than there was MYC prior to the addition of tamoxifen. The true control for this would be to prepare SH-EP-MYCER cells expressed from the same promoter as was MYCNER. Some discussion of qualitative versus quantitative differences should be acknowledged.

Strengths:

Use of a variety of methods to assess the genomic response to increased MYCN in the presence or absence of TFIIIC. Clearly establishes in vitro and in vivo the TFIIIC-MYCN complex

Weaknesses:

Dynamic inferences are made without kinetic experiments.

---

## [Referee Report · Reviewer #3 (Public review)]

Summary:

Vidal et al. investigated how TFIIIC may mediate MYCN effects on transcription. The work builds upon previous reports from the same group where they describe MYCN interactors in neuroblastoma cells (Buchel et al, 2017), which include TFIIIC, and their different roles in MYCN-dependent control of RNA polymerase II function (Herold et al, 2019) (Roeschert et al, 2021) (Papadopoulus et al, 2022). Using baculovirus expression systems, they confirm that MYCN-TFIIIC interaction is direct, and likely relevant for neuroblastoma cell proliferation. However, transcriptomics analyses led them to conclude that TFIIC is largely dispensable for MYCN-dependent gene expression. Instead, they propose that TFIIC limits MYCN-mediated promoter-promoter 3D chromatin contacts, which would in turn facilitate the recruitment of the nascent RNA degradation machinery and restrict the accumulation of non-phosphorylated RNA polymerase II at promoters. How this mechanism may impact on MYCN-driven neuroblastoma cell biology remains to be elucidated.

Strengths:

This study presents a nice variety of genomic datasets addressing the specific role of TFIIIC in MYCN-dependent functions. In particular, the technically challenging HiChIP sequencing experiments performed under various conditions provide very useful information about the interplay between MYCN and TFIIIC in the regulation of 3D chromatin contacts. The authors show that MYCN and TFIIIC participate both in unique and overlapping long-range chromatin contacts and that the expression of each of these proteins limits the function of the other. Together, their results suggest a dynamic and interconnected relationship between MYCN and TFIIIC in regulating 3D chromatin contacts.

Weaknesses:

(1) Mechanistic questions regarding the specific role of TFIIIC in regulating MYCN function remain unsolved. Why is it important to restrict MYCN association to promoter hubs? Do the authors find any TFIIIC-dependent phenotype that is restricted or particularly enhanced at these locations? Both the effects on the accumulation of non-phosphorylated RNA pol II and the recruitment of the nascent RNA degradation machinery seem to be global.

(2) Two specific points regarding RNA pol II ChIPseq results remain unclear:

-It is unfortunate that although both RNAPII (N20) and RNAPII (A10) antibodies were raised against the N-teminal domain, they give different results according to the authors. Caution should be taken, as it may imply that some previous results could be explained by epitope masking.

-I am sorry if I missed something crucial, but to my understanding, the disparities regarding the ChIPseq results obtained using the 8WG16 antibody are not fully resolved. In Figure S7C from their previous publication (Buchel et al, 2017) the authors concluded that "Intriguingly, ChIP sequencing showed that activation of N-MYC had no significant effect on chromatin association of hypo-phosphorylated Pol II". Is this not a similar experiment, using the same antibody and experimental conditions as in Figure 2 from the current manuscript? They now conclude that "activation of MYCN caused a global decrease in promoter association of non-phosphorylated RNAPII".

(3) Conducting ChIP-qPCR experiments for all nascent RNA degradation factors to be compared would have enabled a more direct and comprehensive comparison.

---

## [Author Response]

The following is the authors’ response to the original reviews.

**Public Reviews:**

**Reviewer #1 (Public Review):**
(1) In Figure 1, the authors show that TF3C binds to the amino terminus of MYCN (Myc box I region), as shown previously. The data in Figure 1 B-D support, but do not rigorously confirm a 'direct' interaction because it has not been ruled out that accessory proteins mediating the association may be present in the mixture.

In Figure 1B-D we have purified MYCN and the TFIIIC/TauA complex separately and then mixed the purified preparations, demonstrating that the purified proteins interact. We have additionally performed mass spectrometry, which shows that the TauA/MYCN complex is formed without further accessory proteins, as the molecular weight would be higher. Based on the Coomassie stained SDS-PAGE gels, there is no plausible contaminating band in the purified complex that could be mediating the interaction between MYCN and TauA, either in the purified complex (Figure 1C), or in the purified protein used to reconstitute the complex (Figure S1A & S1B).

(2) The authors indicate in Figure 2 that TF3C has essentially no effect on MYCNdependent gene expression and/or transcription elongation. Yet a previous study (PMID: 29262328) associated with several of the same authors concluded that TF3C positively affects transcription elongation. The authors make no attempt to reconcile these disparate results and need to clarify this point.

We agree that the data in this manuscript do not support the role on transcription elongation. This point was also raised by Reviewer 3. Comparing our new results to the data published previously we can summarize that the data sets in the two studies show three key results: First, the traveling ratio of RNAPII changes upon induction of MYCN. Second, RNAPII decreases at the transcription start side and third, it increases towards the end side.

We agree that in the previous study we linked the traveling ratio directly to elongation. However performing ChIP-seq with different RNAPII antibodies showed us that for example RNAPII (N20), which is unfortunately discontinued, gives different results compared to RNAPII (A10). Combining our new results using the RNAPII (8WG16) antibody shows that the traveling ratio is not only reflecting transcription elongation but also includes that the RNAPII is kicked-off chromatin at the start side.

(3) Figures 2B and C show that unphosphorylated pol2 is TSS-centered, and Ser2-P pol2 occupation is centered beyond the TES. From this data, however, the reader can't tell how much of the phospho-Ser2- pol2 is centered on the TSS. The authors should include overall plots over TSS and TES, and also perhaps the gene-body to allow a better comparison for TSS and TES plotted for both antibodies over the collected gene sets.

We focused on the TSS for unphosphorylated RNAPII and the TES for pSer2-RNAPII, as these are the regions with specific enrichment of the respective antibodies. As requested for comparison, we now include metagenes showing TSS, gene-body, and TES for both antibodies as new Figure S2A and B. Additionally, we included density plots for unphosphorylated RNAPII at the TES as well as for pSer2-RNAPII at the TSS as a Figure for the Reviewers (Figure 1).

(4) The authors see more TF3C at promoters in cells with MYCN (Figure 2F). What are the levels of TF3C in the absence and presence of MYCN?

As shown in the immunoblot in Figure S1E, TF3C5 levels do not change upon induction of MYCN. We therefore think that MYCN helps to recruit TFIIIC5 to RNAPII promoter sites. This is also in accordance to what we previously reported 1.

(5) The finding that TF3C is increased at TSS (Figure 2F) doesn't necessarily indicate that (1) MYCN is recruiting TF3C there, and (2) that this is due to the phosphorylation status of pol2. It could mean many other things. The logic of conflating these 3 points based on the data shown is questionable.

We showed previously that knock-down of MYCN affects TFIIIC5 binding, showing that MYCN is required for binding of TFIIIC5 at promoter sites 1.

Additionally, we included data with DRB treated cells (Figure 2F), which prevents RNAPII loading by preventing downstream de novo elongation. Those data show that TFIIIC5 binding at the TSS is massively increased upon induction of MYCN and additionally upon treatment with DRB. Conversely, we observed that the major effect of TFIIIC knock-down was at the nonphosphorylated RNAPII at the TSS on MYCN induction (Figure 2B). Therefore, we would argue that our assumption fits well to the data presented in the manuscript.

(6) Figure 3A doesn't add much to the paper, as it is overplotted and no relationship is clear, except that Pol2 and MYCN occupy many of the same sites. Perhaps a less complex or different type of plot would allow the interactions to be better visible.

We agree with the comment and since in another comment we were asked to show the same window for all shown Hi-ChIP data plots, we changed Figure 3A.

(7) That depletion of TF3C leads to increased promoter hubs may or may not have anything to do with its association with MYCN (Figure 4E). This could be a direct consequence of its known structural function in cohesin complexes, and the MYCN changes as a secondary consequence of this (also see point 4, above).

As shown in Büchel et al. (2017) 1 MYCN is needed to recruit RAD21 and depletion of RAD21 has no impact on the recruitment of MYCN. Since RAD21 is part of the cohesin complex we would exclude that the MYCN changes are a secondary consequence.

(8) Depletion of TF3C5 results in a loss of EXOSC5 (exosome) at TSS in the presence and absence of MYCN (Figure 5B). As TF3C5 is a cohesin, could this simply be a consequence of genomic structure changes?

We agree that the discovered changes in EXOSC5 can be due to depletion of TFIIIC5. TFIIIC has been shown to recruit cohesin 1 and condensin complexes 2, as well as inducing chromatin architectural changes 3. However, MYCN is needed to recruit TFIIIC and depletion of TFIIIC had no impact on MYCN recruitment 1. Furthermore, MYCN has been shown to recruit exosome 4. Therefore, we would argue that either MYCN can directly play a role or thru chromatin architectural changes.

(9) The authors suggest that RNA dynamics are affected by changes in exosome function (RNA degradation, etc). What effect, if any does TF3C depletion have on the overall gene expression profile?

We show in the manuscript that TFIIIC depletion in unperturbed cells has no effect on the global gene expression profile in the time frame analyzed (Figure 2E and S2B).

**Reviewer #2 (Public Review):**
(1) Dynamic inferences are made without kinetic experiments.

While we agree that we did not collect kinetic data to study the dynamics of RNA polymerase we would argue that the integration of our different data sets make it possible to draw conclusions about dynamic interferences. The transcription cycle and its sequential steps have been well described. In this sense, we use the non-phosphorylated RNAPII data that is situated between RNAPII recruitment and initiation and RNAPII-pSer2 that shows pause-release to elongation to draw conclusions on the dynamic. Likewise, we also made use of our previous published datasets.

**Reviewer #2 (Recommendations For The Authors):**
(1) A number of changes are reported in hub size, expression, etc. upon treatment with tamoxifen to activate MCN-ER. But MYC is already present in the SHEP cells, so why doesn't MYC support these same phenomena? It would seem that either the ability to cooperate with TFIIIC to clear non-productive polymerase complexes from promoters is particular to MYCN, or else it reflects a quantitative increase in total MYC proteins due to the entry of MYCN-ER into the nucleus with tamoxifen. The authors should address or discuss this issue.

It could be that protein levels are the limiting factor between MYC and MYCN observed effects in this system. This interpretation would be in accordance with the results of Lorenzin et al. 5, which reported that different levels of MYC had different targets based on the affinity to Eboxes and protein level. A similar profile of MYC levels compared to function was also reported regarding SPT5 6. Those high protein levels mimic what is found in certain tumors in contrast to physiological levels. In this sense, the observed differences can also be between physiological and oncological levels of MYC proteins.

On the other hand, it has been described both a core MYC- and an isoform specific-signature of target genes. MYCN is described to be involved in gene expression during the S-phase of the cell cycle 7. This suggests that there are differences between MYC and MYCN other than gene sets. The interaction with TFIIIC appears to be one of these differences. We have found multiple TFIIIC subunits as part of the MYCN interactome, but the interaction of TFIIIC with MYC is weaker and we are uncertain how relevant it is 7,8. We show here that depletion of different subunits of the TFIIIC complex show a MYCN-dependent growth defect (Figure 1 E). Similarly, nuclear exosome is a MYCN-specific dependence 4, and we show here that MYCNdependent recruitment of the exosome requires TFIIIC5. We take this as an indication that there is an intrinsic difference between MYC and MYCN and that MYCN engages TFIIIC for this pathway.

(2) Reciprocal to TFIIIC recruitment to MYCN- rRNA, and other RNAPIII genes. Does this happen targets would be MYCN association with tRNA genes, 5S, and if so, is this association TFIIIC dependent? What happens to the expression of these genes?

We did observe MYCN in interactions involving tRNA and other RNAPIII sites, such as SINE elements and tRNAs (Figure 4B, 4D, S3F, and S4B). There was no relevant number of 5S rRNA involved in interactions – either because the difficulty to properly map these repetitive regions or due to biology. In any case, none of those regions appeared to be specifically dependent on TFIIIC as the overall number of interactions increased in TFIIIC depletion regardless of the genomic annotation (Figure S4B). Regarding the expression of RNAPIII genes, we are constrained by technical limitations of poly(A) enrichment RNA-seq to globally analyze it in an unbiased way. However, we addressed this point for tRNAs expression in an earlier work 1 and found that tRNA levels do not change upon TFIIIC depletion. We think this is because tRNAs are stable transcripts and RNAPIII recycling can occur in a TFIIICindependent manner 9. Conversely, we reported no significant expression changes in RNAPII genes upon TFIIIC depletion in this work.

(3) The authors show that TFIIIC depletion does not alter the RNA-expression profile; how do they account for this? Can they comment on "background" transcription that it would seem should be suppressed by TFIIIC-dependent removal of various hypofunctional polymerases?

Since TFIIIC is important for the removal of non-functional RNAPII we would not expect changes to the gene expression profile upon depletion of TFIIIC in the time frame analyzed. Monitoring the elongating form of RNAPII by measuring pSer2 indeed shows us that transcription elongation is not affected.

(4) Global changes in expression are difficult to assess with DESEQ2. This hypernormalizing algorithm is not really suited to distinguish differential, but universal upregulation from some targets being truly upregulated while others are downregulated. The authors should comment.

The authors acknowledge that DESEQ2 relies on the conjecture that genewise estimates of dispersion are generally unchanged among samples. We address this comment in two different ways. We include those in the Figure for the Reviewers (Figure 2). The first was to sequence samples deeper to avoid any bias created by random effect of lower coverage, the range of total reads increased from 6.8-9.3 to 16.5-20.7 million reads. The second was to compare the fold average bin dot plot for RNA-seq of SH-EP-MYCN-ER showing mRNA expression normalized by control per bin using the DESEQ2 (Figure 2A) normalization to TMM in edgeR (Figure 2B) and to quantile normalization (Figure 2C). No major differences were found from the original data or using the different methods, but we updated the Figure 2E in the manuscript to include the deeper sequencing dataset, we also adjusted it to show -/+ MYCN and transformed to log2 to make it more intuitive. Overall, it enhances our original understanding that gene expression remains largely unaffected by TFIIIC5 knockdown.

(5) On page 7, the authors claim that MYCN-ER increased Ser-2 can reflect MYCN-stimulated transcription elongation. In fact, without kinetic studies, this is not fully supported. Accumulation of Ser-2 RNAPII along a gene can reflect increased initiation of full-speed RNAPs or a pile-up of RNAPs slowing down. This should be resolved or qualified.

While we agree that we did not collect kinetic data to study the dynamics of RNA polymerase we would argue that the integration of our different data sets make it possible to draw conclusions about dynamic interferences. We showed on the one side that pSer-2 accumulates on the TES and on the other side the induction of MYCN-ER up-regulates gene expression which proves productive transcription elongation.

(6) pLHiChIP needs to be better described, the Mumbach reference is not sufficient.

We have reformulated the pLHiChIP in the method section and hope that this will provide now a better description of the method.

(7) Can the authors recheck all the labels in Figure 2D-I believe there is an error involving + or - MYCN.

We carefully rechecked all the labels in Figure 2 and it was correct as it was. We understand the confusion that may have created comparing Figure 2D and Figure 2E. To avoid confusion, we updated Figure 2E to show the same direction of Figure 2D. We also log2 transformed the y-axis of Figure 2E to foster a more intuitive reading.

(8) Why are there different scales for the regions of chromosome 17 shown in Figures 3 and 4? It would be easier to compare if the examples were all shown at the same scale (about 2 MB is shown in another Figure).

We now show the same region of chromosome 17 in Figure 3 and 4.

**Reviewer #3 (Public Review):**
(1) The connection between the three major findings presented in this study regarding the role of TFIIIC in the regulation of MYCN function remains unclear. Specifically, how the TFIIICdependent restriction of MYCN localization to promoter hubs enhances the association of factors involved in nascent RNA degradation to prevent the accumulation of inactive RNA polymerase II at promoters is not apparent. As they are currently presented, these findings appear as independent observations. Cross-comparison of the different datasets obtained may provide some insight into addressing this question.

We previously observed that TFIIIC does not affect MYCN recruitment, while MYCN affects TFIIIC binding 1. Moreover, our group reported that MYCN recruits exosome 4 and BRCA1 to promoter-proximal regions 10 to clear out non-functional RNAPII. We are currently reporting that MYCN-TFIIIC complexes exclude non-functional RNAPII. However, MYCN-active promoter hubs have more RNAPII and more transcription than MYCN-active promoter outside hubs. Furthermore, TFIIIC binding occurs upstream of BRCA1 and exosome recruitments as depletion of TFIIIC leads to recruitment decrease of both factors. Therefore, we argue that TFIIIC is required for the proper function of those MYCN-active promoter hubs.

(2) Another concern involves the disparities in RNA polymerase II ChIP-seq results between this study and earlier ones conducted by the same group. In Figure 2, the authors demonstrate that activation of MYCN results in a reduction of non-phosphorylated RNA polymerase II across all expressed genes. This discovery contradicts prior findings obtained using the same methodology, where it was concluded that the expression of MYCN had no significant effect on the chromatin association of hypo-phosphorylated RNA polymerase II (Buchel et al, 2017). In this regard, the choice of the 8WG16 antibody raises concern, as fluctuations in the signal may be attributed to changes in the phosphorylation levels of the Cterminal domain. It remains unclear why the authors decided against using antibodies targeting the N-terminal domain of RNA polymerase II, which are unaffected by phosphorylation and consistently demonstrated a significant signal reduction upon MYCN activation in their previous studies (Buchel et al, 2017) (Herold et al, 2019). Similarly, the authors previously proposed that depletion of TFIIIC5 abrogates the MYCN-dependent increase of Ser2phosphorylated RNA polymerase II (Buchel et al, 2017), whereas they now show that it has no obvious impact. These aspects need clarification.

We politely disagree that our discoveries are contradicting each other. Comparing our new results to the data published previously we can summarize that the data sets in the two studies show three key results: First, the traveling ratio of RNAPII changes upon induction of MYCN. Second, RNAPII decreases at the transcription start side and third, it increases towards the end side.

We agree that in the previous study we linked the traveling ratio directly to elongation. However performing ChIP-seq with different RNAPII antibodies showed us that for example RNAPII (N20), which is unfortunately discontinued, gives different results compared to RNAPII (A10). Combining our new results using the RNAPII (8WG16) antibody shows that the traveling ratio is not only reflecting transcription elongation but also includes that the RNAPII is kicked-off chromatin at the start side.

In the previous study we only performed manual ChIP experiments for RNAPII (8WG16) and pSer2. Now we did a global analysis which is more meaningful and is also reflected in the RNA sequencing data.

(3) Finally, the varied techniques employed to explore the role of TFIIIC in MYCNdependent recruitment of nascent RNA degradation factors make it challenging to draw definitive conclusions about which factor is affected and which one is not. While conducting ChIPseq experiments for all factors may be beyond the scope of this manuscript, incorporating proximity ligation assays (PLA) or ChIP-qPCR assays with each factor would have enabled a more direct and comprehensive comparison.

We understand the criticism that we are comparing different assays. We have performed PLAs with different antibodies. Since the controls of the PLAs were not sufficient for us, we refrain from using them. ChIP-qPCR experiments are much more challenging to do side by side compared to PLAs, which is why we decided against looking at all factors with this method.

**Recommendations For The Authors:**

**Reviewer #3 (Recommendations For The Authors):**
(1) Figure 2: Why did the authors choose the 8WG16 antibody? Does TFIIIC5 depletion suppress the MYCN-dependent reduction of total RNA polymerase II binding to promoters that they consistently showed in previous studies? Given that phosphorylation of the CTD impacts 8WG16 recognition, including Ser5-phosphorylated RNA polymerase II ChIPseq experiments might clarify this issue.

We used the RNAPII (8WG16) antibody to exactly map non-phosphorylated RNAPII which shows us the binding of non-functional RNAPII.

(2) Figures 3 and 4: As it stands, the manuscript does not convincingly establish a functional connection between the results in Figures 2, 3, and 4 or elucidate potential mechanisms. Are changes in RNA polymerase II levels upon MYCN activation more pronounced at promoters located at MYCN hubs? Do changes in MYCN-enriched chromatin contacts upon TFIIIC5 depletion somehow correlate with alterations in RNA polymerase II levels? Performing similar cross-comparisons as in Figure 3C may help address this issue. Furthermore, it not clear how the authors concluded that MYCN/TFIIIC5-bound genes are not part of these so-called promoter hubs.

In Figure 3C we show that RNAPII levels are more pronounced upon MYCN activation at promoters located at MYCN hubs. Additionally, we show non-phosphorylated ChIP-seq on TSS and RNAPII-pSer2 ChIP-seq on TES density plots for promoters with MYCN interactions in the Figure for the Reviewers (Figure 3). We found no other difference than binding compared to the overall global analysis for all expressed genes showed in Figure 2B and Figure 2C. This goes on the same direction of the high expression observed of those genes in MYCN interactions observed in Figure 3C.

The changes observed in Figures 2B and 2C are global and do include the promoters with MYCN interactions. At the same time, it is required a higher number of replicates to statistically distinguish the MYCN interaction differences between TFIIIC5 presence and depletion. We acknowledge this limitation, and we therefore restrain any attempt towards this end. We base our conclusions on the other parts of the manuscript and on our previous studies that show that MYCN recruits TFIIIC, BRCA1, and the exosome to promoter proximal regions 1,4,10.

(3) Figure 5: According to the PLA results, activation of MYCN could enhance RNA polymerase II-NELFE interaction in a TFIIC5-dependent manner. Considering the raised issues regarding the use of the 8WG16 antibody, this result might be of relevance.Nevertheless, PLA does not seem to be the optimal technique to address these questions, and I would rather suggest performing ChIP-qPCR experiments for all the factors to be compared. Finally, do the authors conclude that the TFIIIC5 effect on MYCN-dependent changes in RNA polymerase II depends upon the recruitment of EXOSC5 and BRCA1? If so, it would be interesting to determine whether depletion of these factors phenocopies the effects observed with TFIIC5.

We understand the criticism that we are comparing different assays. We have performed PLAs with different antibodies. Since the controls of the PLAs were not sufficient for us, we refrain from using them.

(4) In Figure S2 the labels should be EtOH, 4-OHT, and Input.

We changed this accordingly.

(5) On page 7, the sentence "We have shown previously that TFIIIC5 depletion does not cause significant changes in expression of multiple tRNA genes that are transcribed by RNAPIII (Buchel et al., 2017)" appears to lack a connection.

We agree with the reviewer and we deleted this sentence from the manuscript.

**Author response image 1. sa4fig1:** (A) Density plot of ChIP-Rx signal for non-phosphorylated RNAPII. Data show mean (line) ± standard error of the mean (SEM indicated by the shade) of different gene sets based on an RNA-seq of SH-EP-MYCN-ER cells ± 4-OHT. The y-axis shows the number of spike-in normalized reads and it is centred to the TES ± 2 kb. N = number of genes in the gene set defined in the methods. (B) Density plot of ChIP-Rx signal for RNAPII pSer2 as described for panel A. The signal is centred to the TSS ± 2 kb.

**Author response image 2. sa4fig2:** Bin dot plot for RNA-seq of SH-EP-MYCN-ER showing mRNA expression normalized by control per bin comparing the fold average using DESEQ2 (A), normalization to TMM in edgeR (B) and to quantile normalization (C).

**Author response image 3. sa4fig3:** Average density plot of ChIP-Rx signal for non-phosphorylated RNAPII (A) or RNAPII pSer2 (B) at promoters with MYCN interactions.

References

(1) Büchel, G., Carstensen, A., Mak, K.-Y., Roeschert, I., Leen, E., Sumara, O., Hofstetter, J., Herold, S., Kalb, J., and Baluapuri, A. (2017). Association with Aurora-A controls NMYC-dependent promoter escape and pause release of RNA polymerase II during the cell cycle. Cell reports 21, 3483-3497.

(2) Yuen, K.C., Slaughter, B.D., and Gerton, J.L. (2017). Condensin II is anchored by TFIIIC and H3K4me3 in the mammalian genome and supports the expression of active dense gene clusters. Sci Adv 3, e1700191. 10.1126/sciadv.1700191.

(3) Ferrari, R., de Llobet Cucalon, L.I., Di Vona, C., Le Dilly, F., Vidal, E., Lioutas, A., Oliete, J.Q., Jochem, L., Cutts, E., Dieci, G., et al. (2020). TFIIIC Binding to Alu Elements Controls Gene Expression via Chromatin Looping and Histone Acetylation. Mol Cell 77, 475-487 e411. 10.1016/j.molcel.2019.10.020.

(4) Papadopoulos, D., Solvie, D., Baluapuri, A., Endres, T., Ha, S.A., Herold, S., Kalb, J., Giansanti, C., Schulein-Volk, C., Ade, C.P., et al. (2021). MYCN recruits the nuclear exosome complex to RNA polymerase II to prevent transcription-replication conflicts. Mol Cell. 10.1016/j.molcel.2021.11.002.

(5) Lorenzin, F., Benary, U., Baluapuri, A., Walz, S., Jung, L.A., von Eyss, B., Kisker, C., Wolf, J., Eilers, M., and Wolf, E. (2016). Different promoter affinities account for specificity in MYC-dependent gene regulation. Elife 5. 10.7554/eLife.15161.

(6) Baluapuri, A., Hofstetter, J., Dudvarski Stankovic, N., Endres, T., Bhandare, P., Vos, S.M., Adhikari, B., Schwarz, J.D., Narain, A., Vogt, M., et al. (2019). MYC Recruits SPT5 to RNA Polymerase II to Promote Processive Transcription Elongation. Mol Cell 74, 674-687 e611. 10.1016/j.molcel.2019.02.031.

(7) Baluapuri, A., Wolf, E., and Eilers, M. (2020). Target gene-independent functions of MYC oncoproteins. Nat Rev Mol Cell Biol. 10.1038/s41580-020-0215-2.

(8) Koch, H.B., Zhang, R., Verdoodt, B., Bailey, A., Zhang, C.D., Yates, J.R., 3rd, Menssen, A., and Hermeking, H. (2007). Large-scale identification of c-MYCassociated proteins using a combined TAP/MudPIT approach. Cell Cycle 6, 205-217. 10.4161/cc.6.2.3742.

(9) Ferrari, R., Rivetti, C., Acker, J., and Dieci, G. (2004). Distinct roles of transcription factors TFIIIB and TFIIIC in RNA polymerase III transcription reinitiation. Proc Natl Acad Sci U S A 101, 13442-13447. 10.1073/pnas.0403851101.

(10) Herold, S., Kalb, J., Büchel, G., Ade, C.P., Baluapuri, A., Xu, J., Koster, J., Solvie, D., Carstensen, A., and Klotz, C. (2019). Recruitment of BRCA1 limits MYCN-driven accumulation of stalled RNA polymerase. Nature 567, 545-549.